# Plant Natural Products for the Control of *Aedes aegypti*: The Main Vector of Important Arboviruses

**DOI:** 10.3390/molecules25153484

**Published:** 2020-07-31

**Authors:** Maíra Rosato Silveiral Silvério, Laila Salmen Espindola, Norberto Peporine Lopes, Paulo Cézar Vieira

**Affiliations:** 1NPPNS, Departamento de Ciências BioMoleculares, Faculdade de Ciências Farmacêuticas de Ribeirão Preto, Universidade de São Paulo, Ribeirão Preto 14040-903, São Paulo, Brazil; mairarosato@gmail.com; 2Laboratório de Farmacognosia, Universidade de Brasília, Brasília 70910-900, Brazil; darvenne@unb.br

**Keywords:** *Aedes aegypti*, dengue, natural products, botanical species, essential oils, terpenes, phenylpropanoids, thiophenes, alkaloids, mechanisms of action

## Abstract

The mosquito species *Aedes aegypti* is one of the main vectors of arboviruses, including dengue, Zika and chikungunya. Considering the deficiency or absence of vaccines to prevent these diseases, vector control remains an important strategy. The use of plant natural product-based insecticides constitutes an alternative to chemical insecticides as they are degraded more easily and are less harmful to the environment, not to mention their lower toxicity to non-target insects. This review details plant species and their secondary metabolites that have demonstrated insecticidal properties (ovicidal, larvicidal, pupicidal, adulticidal, repellent and ovipositional effects) against the mosquito, together with their mechanisms of action. In particular, essential oils and some of their chemical constituents such as terpenoids and phenylpropanoids offer distinct advantages. Thiophenes, amides and alkaloids also possess high larvicidal and adulticidal activities, adding to the wealth of plant natural products with potential in vector control applications.

## 1. Introduction

The mosquito *Aedes aegypti* (Diptera: Culicidae) originated in Egypt and it is widely distributed in tropical and subtropical regions, including North America and Europe [1,2]. *Ae. aegypti* presents complete metamorphosis from immature egg, larva and pupa stages to the adult mosquito itself (Figure 1). The life cycle varies according to environmental temperature, food availability and quantity of larvae in the same breeding site. Under favorable conditions, after egg hatching, the mosquito transforms into the adult stage within 10 days, even though the eggs can be viable up to 450 days in the absence of water [3].

The female mosquito requires hematophagy for egg maturation. Viral transmission to humans occurs during this process if the mosquito is infected. The lifetime of a female mosquito is approximately 45 days [1]. *Ae. aegypti* population control is considered the principal measure to combat arboviral diseases as this species is the primary vector of dengue, Zika, chikungunya and urban yellow fever [4,5].

In 2012, dengue was considered the mosquito-borne disease of major importance in the world [5]. According to the World Health Organization, 390 million people are infected annually with the dengue virus, 96 million of which have clinical manifestations [6]. There are various symptoms, the first is usually high fever (39–40 °C) with headache, prostration, arthralgia, anorexia, asthenia, nausea, among others. Some clinical aspects often depend on patient age. There is no specific treatment for dengue and the more complicated cases of the disease can cause hemorrhage, shock and even death [7].

*Ae. aegypti* is considered the main Zika virus vector (Figure 1), but infection can also occur by sexual transmission or blood transfusion [8]. Symptoms are non-specific and self-limited, being easily confused with other arboviral diseases. Some important complications exist, such as microcephaly in fetuses and Guillan Barré syndrome. The virus has been reported in countries in Americas, Europe, Asia and the Pacific region [8]. Like dengue and Zika, chikungunya has no specific treatment. The disease emerged in the Americas in 2013, with about 1.7 million cases identified and 252 deaths reported by August 2015 [9].

In 2013, dengue generated a global cost of US $8.9 billion, with around 58.4 million symptomatic cases (13.5 million fatalities) in the 141 countries and territories. The per capita costs of dengue were $70.1 for hospital treatment, $51.1 for outpatient treatment and $12.9 for cases that did not reach the health system. According to this study, Brazil had an incidence of 751 to 1,000 cases per 100,000 people. The expenses were proportional to the incidence and ranged from $2.5 to $5 for each treated case [10].

A more recent study showed that in 2016 the Brazilian government spent around R $805 million (ca. 160 million US$) to treat diseases caused by the *Ae. aegypti* mosquito, including direct medical expenses and indirect costs. In addition, about R $1.5 billion were destined to combat the vector, totaling R $2.3 billion, that was 2% of the health budget for that year. More than 2 million cases of *Ae. aegypti* related diseases were verified. These numbers were underestimated as they did not include complications such as microcephaly and Guillain-Barré syndrome [11].

A study estimated that about 60% of the world population will be at risk of dengue in 2080, which represents over 6.1 (4.7–6.9) billion people [12]. Considering that vector control is the main tool for controlling these expensive arboviruses, investment in techniques to combat the *Ae. aegypti* mosquito is growing [3]. Investments are particularly focused on techniques with minimal negative impacts on non-target animals and the environment [4,13].

## 2. Mosquito Control

There are several techniques already used to combat mosquitoes, which act both in the immature phases (egg, larva and pupa) and in the adult [13]. Highly toxic synthetic insecticides such as organophosphates, pyrethroids and carbamates have been historically used to combat the mosquito, acting mainly on insect larvae [4]. More recently, insecticides with less toxicity which are less persistent in the environment have been developed, including neonicotinoids and oxadiazines [14]. However, these products are still widely used and harmful to living organisms and the environment, and the use of foogers and aerial applications of sintetic insecticides against adults, such as pyrethroids products, contributes to insect resistance problems [4]. Therefore, efforts must be made to ensure newly developed alternative insecticides are more eco-friendly.

Biological tools that control the adult stage are based on behavior, such as the Sterile Insect Technique (SIT), Incompatible Insect Technique (IIT) and Release of Insects carrying a Dominant Lethal gene (RIDL), which involve insect sterilization by chemical irradiation, natural bacteria which are pathogenic for mosquito (highly specific strains of *Wolbachia*) and genetic modifications to make sterile male mosquitoes, respectively [15,16]. The other technique applied against the adult stage is the use of entomopathogenic fungi, specifically in the orders *Entomophthorales*, *Hypocreales* and *Pezizales* due to their specificity, ability to manipulate and infectiveness to the host [17].

*Ae. aegypti* control using specific strains of *Wolbachia* bacteria is currently practiced in different locations around the world through the World Mosquito Program. This program involves the application of the bacteria to laboratory mosquitoes that are released into the local *Ae. aegypti* population during reproduction. The presence of bacteria in mosquitoes decreases the possibility of arbovirus transmission to people [18].

Biological control tools that act against the immature stages include the application of *Bacillus thuringiensis* in larvae habitats; products that prevent oviposition and/or inhibit growth and reproduction, including pheromones. There are also natural predators such as fish (especially of the genus *Gambusia* and *Poecilia*, family Poeciliidae) [19,20], copepods (including several species of the genus *Mesocyclops*) [21,22] and the “elephant mosquito” (genus *Toxorhynchites*) [23,24].

Finally, plant-based insecticides (ovicides, larvicides and pupicides) [25,26] deserve a special mention due to the vast biodiversity of species found in the world, estimated to be approximately 400,000 terrestrial species [27]. Botanical insecticides can be plant extracts, essential oils and/or secondary metabolites [4,14].

## 3. Plant Natural Products to Control Mosquitoes

The search for plant natural products to control *Ae. aegypti* dates back a number of years, with research published since the 1980s [28,29]. However, chemical insecticides are most commonly used, despite their enormous toxicity to non-target organisms, such as: (i) poisoning and death; (ii) cancer, by non-genotoxic mechanisms (immunosuppressants, cytotoxic) or by triggering the carcinogenic process in different ways; (iii) harmful effects on the nervous, renal, respiratory and reproductive systems and (iv) induction of oxidative stress [30].

In addition to toxicity, another concern is the increasing resistance of the mosquito vector to chemical insecticides. One example is the knockdown resistance (*kdr*) mutation, in which resistance to pyrethroid insecticides occurs, whereby the target site is the sodium channel of the *Ae. aegypti* nervous system [31,32]. In Brazil, of the five insecticides approved by the Public Health Ministry and recommended by the WHO for adult mosquito control, four belong to the pyrethroid class together with one organophosphate (malathion). However, in 2011 a technical note was issued suspending the use of pyrethroids in Brazil to control *Ae. aegypti* [33].

The level of resistance is dependent on the insecticide concentration, frequency and duration of application [34]. The resistance mechanisms of mosquitoes may be associated with changes in the insect cuticle resulting in less insecticide absorption [35], changes in insect metabolism involving biotransformation enzymes [36,37] and modifications of the insecticide target site, usually by genetic mutations [38,39].

The main esterases involved in the resistance process are carboxylesterases and cholinesterases. Carboxylesterases are usually resistant to organophosphates, with this resistance relating to both a quantitative mechanism (overproduction of enzyme) and qualitative mechanism (mutations that cause alterations in enzymatic properties) [40]. In the case of cholinesterases, the resistance is mainly caused by gene mutation. The main insecticides resistant to the acetylcholinesterase target site are organophosphates and carbamates [41].

Another problem associated with chemical insecticides is the damage caused to the environment and living organisms by their degradation products, which may prove more toxic than the original product itself. Examples include the degradation products of temephos, whose effects have already been documented in aquatic environments [42], and malathion, together with its metabolites, in non-target organisms such as *Daphnia magna* (Cladocera: Daphniidae) [43].

Insecticides derived from plant natural products therefore offer a promising source of safer new products for mosquito control due to minimal residues from its natural degradation in both the field and in water, minimizing ecosystem disruption [44,45]. There is considerable research on insecticides of natural origin, especially those of microbial and plant origin, due to their innumerable secondary metabolites produced especially as a defense mechanism against natural predators [46]. It is estimated that there are more than 100,000 plant metabolites, with hundreds or more exhibiting some activity against insects [47].

Botanical insecticides are advantageous as they are generally environmentally safe, non-toxic to non-target organisms including homeothermic animals and their residues biodegradable [25,26,30]. The synergic mixture of the active compounds in extracts induce several mechanisms of action and result in less pest resistance [30,48].

The present review focuses on the more recent studies of botanical extracts and active compounds in applications against *Ae. aegypti*, from immature to adult stages, in addition to their main proposed mechanisms of action. The crude extracts are obtained using different extraction methods with organic solvents or water. Essential oils are obtained by steam distillation or hydrodistillation. The classes of active compounds include terpenes, alkaloids and amides, steroids, flavonoids, furanochromones, phenylpropanoids and phenol derivatives, lignans and neolignans, naphthoquinones, fatty acids and their derivatives. The type of insecticide activity (ovicide, larvicide, pupicide, adulticide) is reported as mortality and lethal concentration values (LC_50_, LC_90_ and/or LC_99_), together with egg hatchability. The other activities tested are mosquito repellency, oviposition deterrence, growth regulation and the antifeedant effect.

## 4. Essential Oils

Essential oils deserve special attention as they have yields of 0.5 to 2.0% in the extraction process, contain a high concentratration of secondary metabolites and generally present potent activity due to the synergic effect of the constituents. An important advantage is, with few exceptions, their relatively low, or no, toxicity to mammals (Figure 2). Some pure compounds constituents of essential oils are moderately toxic to mammals (LD_50_ 800–3000 mg/kg in rodents) while formulated products usually are low or non-toxic to mammals, birds and fish (LD_50_ above 5000 mg/kg for rodents) [47,49].

These essential oils are mainly obtained from aromatic plants, of which there are more than 3000 species. Approximately 10% of these are already produced in large quantities for other uses, such as flavorings and fragrances, and are therefore readily available at reasonable prices [30]. Essential oils are composed of volatile compounds, which give an important advantage of non-persistence in the environment [49,50].

It is important to note that the same volatility may be a disadvantage in terms of instability. However, this property can be overcome using pharmaceutical technology such as micro and nanoencapsulation [51,52]. Formulation development is therefore critical for essential oils to be used effectively and safely as pesticides. A number of studies have demonstrated that a suitable vehicle prolongs the insecticidal effect [51,53,54].

Of the plant families affording essentials oils, those most tested against *Ae. aegypti* larvae were Myrtaceae, in particular *Eucalyptus* species, followed by Fabaceae, Asteraceae, Apiaceae and Lamiaceae. Asteraceae was the most important for the adulticide, repellent and oviposition effects.

Regarding larvicidal activity there is currently no value specified by the WHO to discriminate whether a compound or extract is active against insects. However, researchers usually consider that an LC_50_ < 50 µg/mL is very active; an LC_50_ 50–100 µg/mL is active, and an LC_50_ > 100 µg/mL is weak/inactive [55,56,57].

Considering this classification, this review highlights 11 species with 12 very active essential oils, 11 species with 14 active essential oils and 6 species with weak/no activity for 7 essential oils. Eight species do not have reported LC_50_ values and are not considered in this classification. However, these values can change significantly after formulation, as discussed in Section 8 “Limitations and/or Expectations of Plant Natural Product Insecticide Applications”.

Some studies made the identification of secondary metabolites in essential oils evaluated for insectidal activities described abouve and its chemical structures are illustrated in Figure 3.

Table 1 and Table 2 summarise the publications selected for this review and discussion of the essential oils active against the *Ae. aeygpti* mosquito. Table 1 describes larvicidal activities, while Table 2 details the adulticidal, repellent and oviposition activities.

The essential oil from *Acacia nilotica* (L.) Delile (Fabaceae) seeds had the highest insecticidal activity (LC_50_ 3.17 µg/mL). The major compounds were hexadecane (**1**) and heptacosane (**2**) [58]. This species has been investigated for several medicinal uses including spasmogenic and antiplasmodial activities of seed extracts [59].

Another species is *Myristica fragans* Houtt. (Myristicaceae), which is popularly known as nutmeg and is used as a flavoring. Essential oil from its seeds demonstrated high toxicity against *Ae. aegypti*, in both the L3 larval phase (LC_50_ 28.2 µg/mL) and the adult phase (LC_50_ 18.5 µg/mg female). The major compounds identified were sabinene (**3**, 52%), α-pinene (no stereochemistry defined, **4**) (13%) and terpinen-4-ol (**5**) (11%). Regarding neurotoxic effects, this essential oil is non-toxic to humans as its IC_50_ values for human acetylcholinesterase and human butyrylcholinesterase are higher than 4000 µg/mL [54]. Nutmeg flower essential oil presented higher larvicidal activity (LC_50_ 47.42 μg/mL) than the ethanolic extract (LC_50_ 75.45 μg/mL). This result suggests that the constituents of the essential oil either exhibit higher larvicidal activity, or that the synergy between them favors the toxicity to the mosquito [60].

*Eucalyptus* species (Myrtaceae) leaf essential oils showed LC_50_ values in the range of 31.0–95.5 µg/mL for the larvae stage and 100% repellency for 1.5 h. *Eucalyptus camaldulensis* Dehnh. had the highest larvicidal activity while *Eucalyptus saligna* Sm. displayed higher repellency than *Eucalyptus nitens* (H. Deane & Maiden) Maiden [61,62,63].

*Cinnamomum osmophloeum* Kaneh. (Lauraceae) is commonly named pseudocinnamomum and the essential oil from leaves of different places demonstrating larvicidal activities of LC_50_ 36 to 177 µg/mL [64]. Similar variation was observed in the larvicidal activity of different guava crops. The LC_50_ values of *Psidium guajava* L. (Myrtaceae) leaf essential oil ranged from 39.48 to 64.25 µg/mL [65].

Other edible plant essential oils that showed strong larvicidal activity were parsley [*Petroselinum crispum* (Mill.) A.W. Hill (Apiaceae)], fennel [*Foeniculum vulgare* Mill. (Apiaceae)], star anise [*Illicium verum* Hook. f. (Illiciaceae)], *Piper sarmentosum* Roxb. ex Hunt. (Piperaceae) and sucupira [*Pterodon emarginatus* Vogel (Fabaceae)] [54,60,66].

For parsley, the toxicity was evaluated for pyrethroid-susceptible and pyrethroid-resistant mosquitoes (LC_50_ 40 µg/mL) and the major metabolite was thymol (**6**). Similar toxicity was observed for fennel (LC_50_ 44.84 µg/mL), for star anise (LC_50_ 39.8 µg/mL) and for *Piper sarmentosum* (LC_50_ 49.19 µg/mL) [54,60]. The major constituent of star anise essential oil was *trans*-anethole (**7**, 90%) [54]. Furthermore, parsley and star anise essential oils demonstrated adulticidal activity with LC_50_ 6.01 µg/mg female for pyrethroid-susceptible, LC_50_ 6.15 µg/mg female for pyrethroid-resistant (*Petroselinum crispum*) and LC_50_ 10.3 µg/mg female for *Illicium verum* [54,60].

A sucupira (*Pterodon emarginatus*) fruit essential oil incorporated into a nanoemulsion to improve water solubility and increase product stability presented an LC_50_ 34.75 µg/mL. The metabolites were identified as β-caryophyllene (**8**), geranylgeraniol (**9**) and 6α,7β-dihydroxyvouacapan-17-β-oic acid (**10**). The toxicity for non-target organisms was tested in adult female Swiss albino mice [*Mus musculus* (Rodentia: Muridae)] with no behavioral effects, macroscopical changes or deaths reported [66].

**Table 1 molecules-25-03484-t001:** Larvicidal activity of essential oils against the *Ae. aegypti* mosquito.

Plant Species	Family	Country	Part Used	Larval Stage	Mortality				Time of Analysis (h)	Reference
					% Death	Concentration (ppm)	LC_50_ (ppm)	LC_90_ (ppm)		
*Acacia nilótica* (L.) Delile	Fabaceae	India	Seeds	L4	ND	ND	3.17	11.73	24	[58]
*Alpinia purpurata* (Viell.) K. Schum.	Zingiberaceae	Brazil	Red flowers	L4	ND	ND	80.70	ND	24	[67]
			Pink flowers	L4	ND	ND	71.50			
*Baccharis reticularia* DC.	Asteraceae	Brazil	Leaves	L4	ND	ND	221.27	457.47	24	[68]
*Bauhinia pulchella* Benth.	Fabaceae	Brazil	Leaves	L3	ND	ND	105.90	ND	24	[69]
*Bauhinia ungulata* L.	Fabaceae	Brazil	Leaves	L3	ND	ND	75.10	ND	24	[69]
*Cinnamomum osmophloeum* Kaneh.	Lauraceae	Taiwan	Leaves	L4	ND	ND	36.0 to 177.0	79.0 to 296.0	24	[64]
*Croton rhamnifolioides* Pax & K. Hoffm.	Euphorbiaceae	Brazil	Leaves	L4	ND	ND	89.0 and 122.30	ND	24	[70]
*Cunninghamia konishii* Hayata	Taxodiaceae	Taiwan	Wood	L4	ND	ND	85.70	171.40	24	[71]
			Leaves	L4	ND	ND	91.70	176.50		
*Curcuma longa* L.	Zingiberaceae	Thailand	Rhizome	L4 (p-s)	ND	ND	65.51	110.93	24	[60]
*Eucalyptus camaldulensis* Dehnh.	Myrtaceae	Taiwan	Leaves	L4	ND	ND	31.0	71.80	24	[61]
*Eucalyptus nitens* (H. Deane & Maiden) Maiden	Myrtaceae	Argentina	Leaves	L3/L4	ND	ND	52.83	ND	24	[62]
*Eucalyptus urophylla* S.T. Blake	Myrtaceae	Taiwan	Leaves	L4	ND	ND	95.50	166.30	24	[61]
*Ferula galbaniflua* Boiss. & Buhse	Apiaceae	Corea	Resin	L3	90	100.0	ND	ND	48	[45]
*Foeniculum vulgare* Mill.	Apiaceae	Thailand	Fruit	L4 (p-s)	ND	ND	44.84	57.05	24	[60]
*Hyssopus officinalis* L.	Lamiaceae	Corea	Flowers	L3	95	100.0	ND	ND	48	[45]
*Illicium verum* Hook. f.	Illiciaceae	Brazil	Fruit	L3	ND	ND	39.80	53.0	24	[54]
*Larix europea* Lam. & A. DC.	Pinaceae	Corea	Resin	L3	87	100.0	ND	ND	48	[45]
*Limnophila aromatica* (Lamk.) Merr.	Scrophulariaceae	Thailand	Whole plant	L4 (p-s)	ND	ND	47.94	65.14	24	[60]
*Mentha spicata* L.	Lamiaceae	India	Leaves	L3	ND	ND	56.08	110.28	24	[72]
*Myristica fragrans* Houtt.	Myristicaceae	Thailand	Flowers	L4 (p-s)	ND	ND	47.42	69.28	24	[60]
		Brazil	Seeds	L3	ND	ND	28.20	41.70	24	[54]
*Myroxylon pereirae* (Royle) Klotzsch	Fabaceae	Corea	Resin	L3	97.5	100.0	ND	ND	48	[45]
					95	50.0	ND	ND		
*Pelargonium graveolens* L’Hér. ex Aiton	Geraniaceae	Corea	Leaves	L3	82	100.0	ND	ND	48	[45]
*Petroselinum crispum* (Mill.) A.W. Hill	Apiaceae	Thailand	Fruit	L4 (p-s)	ND	ND	43.22	66.60	24	[60]
				L4 (p-s 1)	ND	ND	44.50	68.29	24	
				L4 (p-s 2)	ND	ND	44.03	67.71	24	
*Pimenta dioica* (L.) Merr.	Myrtaceae	Brazil	Fruit	L3	ND	ND	104.40	137.30	24	[54]
*Pinus sylvestris* L.	Pinaceae	Nigeria	Needles	L4	ND	ND	100.39	ND	24	[73]
*Piper aduncum* L.	Piperaceae	Brazil	Leaves	L3/L4	ND	ND	289.90	654.90	24	[74]
					ND	ND	134.10	527.10	48	
*Piper sarmentosum* Roxb. ex Hunt.	Piperaceae	Thailand	Stem and Leaves	L4 (p-s)	ND	ND	49.19	75.10	24	[60]
*Pogostemon patchouli* Pellet.	Lamiaceae	Corea	Whole plant	L3	97	100.0	ND	ND	48	[45]
*Porophyllum ruderale* (Jacq.) Cass.	Asteraceae	Brazil	Flowers and leaves	L3	ND	ND	60.90	132.48	24	[75]
				L4	ND	ND	72.28	173.65	24	
*Psidium guajava* L.	Myrtaceae	Brazil	Leaves	L4	ND	ND	39.48 to 64.25	57.34 to 86.0	24	[65]
*Pterodon emarginatus* Vogel	Fabaceae	Brazil	Fruit	L4	ND	ND	34.75	ND	48	[66]
*Rosmarinus officinalis* L.	Lamiaceae	Brazil	Leaves	L4	80	250.0	ND	ND	24	[76]
					90	250.0	ND	ND	48	
*Sphaeranthus indicus* L.	Asteraceae	India	Leaves	L4	ND	ND	140.0	350.0	24	[77]
*Syzygium aromaticum* (L.) Merr. & L.M. Perry	Myrtaceae	Nigeria	Bud	L4	ND	ND	92.56	ND	24	[73]
*Tanacetum argenteum* (Lam.) Willd. subsp. argenteum (Lam.)	Asteraceae	Turkey	Aerial parts	L1	ND	ND	93.30	241.70	24	[78]
*Trachyspermum ammi* (L.) Sprague	Apiaceae	Corea	Seeds	L3	100	100.0	ND	ND	48	[45]
					80	50.0	ND	ND		

LC_50_ lethal concentration required to kill 50% of the larval population, LC_90_ lethal concentration required to kill 90% of the larval population, ND not described, p-s pyrethroid-susceptible.

**Table 2 molecules-25-03484-t002:** Adulticidal, repellent and oviposition activities of essential oils against the *Ae. aegypti* mosquito.

Plant Species	Family	Country	Part Used	Activity	Results	Time of Analysis (h)	Reference
*Acantholippia seriphioides* (A. Gray) Moldenke	Verbenaceae	Argentina	ND	Repellent	100% of repellency at 50%	1.2	[63]
*Aloysia citriodora* Palau	Verbenaceae	Argentina	ND	Repellent	100% of repellency at 12.5%	1.5	[63]
*Alpinia purpurata* (Viell.) K. Schum.	Zingiberaceae	Brazil	Red flowers	Oviposition	Oviposition disruptive effect	ND	[67]
			Pink flowers	Oviposition	Oviposition disruptive effect	ND	[67]
*Baccharis spartioides* (Hook. & Arn.) Remy	Asteraceae	Argentina	ND	Repellent	100% of repellency at 12.5%	1.5	[63]
*Croton rhamnifolioides* Pax & K. Hoffm.	Euphorbiaceae	Brazil	Leaves	Oviposition	Only 30% of oviposition at 100.0 µg/mL	16	[70]
*Eucalyptus nitens* (H. Deane & Maiden) Maiden	Myrtaceae	Argentina	Leaves	Repellent	100% pure repellency	1.5	[62]
*Eucalyptus saligna* Sm.	Myrtaceae	Argentina	ND	Repellent	100% of repellency at 50%	1.5	[63]
*Illicium verum* Hook. f.	Illiciaceae	Brazil	Fruit	Adulticide	LC_50_ 10.30 µg/mg female LC_90_ 17.50 µg/mg female	24	[54]
*Minthostachys mollis* Griseb	Lamiaceae	Argentina	ND	Repellent	100% of repellency at 50%	1.0	[63]
*Myristica fragrans* Houtt.	Myristicaceae	Brazil	Seeds	Adulticide	LC_50_ 18.50 µg/mg female LC_90_ 31.90 µg/mg female	24	[54]
*Petroselinum crispum* (Mill.) A.W. Hill	Apiaceae	Thailand	Fruit	Adulticide (p-s)	LC_50_ 6.01 µg/mg female LC_90_ 9.39 µg/mg female	24	[60]
				Adulticide (p-r)	LC_50_ 6.15 µg/mg female LC_90_ 9.82 µg/mg female	24	[60]
*Pimenta dioica* (L.) Merr.	Myrtaceae	Brazil	Fruit	Adulticide	LC_50_ 16.6 µg/mg female LC_90_ 31.4 µg/mg female	24	[54]
*Pluchea carolinensis* (Jack.) G.	Asteraceae	Martinica	Leaves and flowers	Repellent	36.6% of repellency at 1.0%	ND	[79]
				Irritant	66.2% of irritation at 0.1%	ND	[79]
*Rosmarinus officinalis* L.	Lamiaceae	Argentina	ND	Repellent	100% of repellency at 50%	1.5	[63]
*Sphaeranthus indicus* L.	Asteraceae	India	Leaves	Repellent	100% of repellency at 200 ppm	3.5	[77]
				Adulticide	100% of mortality at 800 ppm	24	[77]
*Tagetes minuta* L.	Asteraceae	Argentina	ND	Repellent	100% of repellency at 25%	1.5	[63]

LC_50_ lethal concentration required to kill 50% of the mosquito population, LC_90_ lethal concentration required to kill 90% of the mosquito population, ND not described, p-s pyrethroid-susceptible, p-r pyrethroid-resistant.

The essential oil of spearmint [*Mentha spicata* L. (Lamiaceae)] leaves also showed larvicidal activity (LC_50_ 56.08 µg/mL). The main constituents were carvone (**11**) (48.6%), *cis*-carveol (**12**, 21.3%) and limonene (**13**, 11.3%) [72]. Another spice with larvicidal acitivity was turmeric [*Curcuma longa* L. (Zingiberaceae)] whose rhizome essential oil demonstrated LC_50_ 65.51 µg/mL [62]. Similarly, *Porophyllum ruderale* (Jacq.) Cass. (Asteraceae) leaves essential oil, a herb used for seasoning food, showed LC_50_ 60.9 µg/mL for L3 larvae and LC_50_ 72.3 µg/mL for L4. The main metabolite identified was β-ocimene (**14**, 94%) [75].

Two other species exhibited very similar LC_50_ values (ca. 93 µg/mL): clove [*Syzygium aromaticum* (L.) Merr. & L.M. Perry (Myrtaceae)] against L4 larvae and *Tanacetum argenteum* (Lam.) Willd. subsp. argenteum (Lam.) (Asteraceae) against L1 [73,78]. The major constituent of clove bud essential oil was eugenol **15** (80%) [73]. The clove bud alcoholic extract is popularly used as a repellent.

Cheng et al. (2013) demonstrated that essential oils of different parts of *Cunninghamia konishii* Hayata (Taxodiaceae) were toxic to L4 larvae (wood LC_50_ 85.7 µg/mL) and (leaves LC_50_ 91.7 µg/mL). The ethanolic extracts were inactive: wood (LC_50_ 240 µg/mL) and leaves (LC_50_ > 400 µg/mL). The essential oil major compounds were: cedrol (**16**, 53.0%) and α-pinene (**4**, 25.6%, wood), and α-pinene (**4**, 35.9%) and *p*-cymene (**17**, 16.7%, leaves) [71]. The results suggest that compound synergy is important for essential oil larvicidal activity as isolated compounds showed lower toxicity, with the exception of *p*-cymene [71].

*Alpinia purpurata* (Viell.) K. Schum. (Zingiberaceae) and *Croton rhamnifolioides* Pax & K. Hoffm. (Euphorbiaceae) demonstrated toxicity against L4 larvae and oviposition deterrent effect. The LC_50_ of *A. purpurata* essential oil was 71.5 µg/mL (pink flowers) and 80.7 µg/mL (red flowers) [67], while the LC_50_ for *C. rhamnifolioides* was 89.0 µg/mL (fresh leaves) and 122.3 µg/mL (stored leaves) [70].

Other species, such as *Baccharis reticularia* DC. (Asteraceae); *Piper aduncum* L. (Piperaceae) and *Pinus sylvestris* L. (Pinaceae) showed weak larvicidal activities (LC_50_ 100.4–290.0 µg/mL) [68,73,74]. *Bauhinia pulchella* Benth. (Fabaceae) displayed weak activity (105.9 µg/mL) whereas *Bauhinia ungulata* L. (Fabaceae) was active (75.1 µg/mL) [69] and *Pimenta dioica* (L.) Merr. (Myrtaceae) was investigated for both larvicidal (weak activity, LC_50_ 104.4 µg/mL) [54] and adulticidal action (very active, LC_50_ 16.6 µg/mL) [24].

The essential oil of *Sphaeranthus indicus* L. (Asteraceae) leaves showed poor activity: larvae (LC_50_ 140 µg/mL), adult (800 µg/mL for 100% mortality) and repellent (200 µg/mL). This essential oil showed low toxicity (1500 µg/mL) to a non-target aquatic predator [*Toxorhynchites splendens* (Diptera: Culicidae)] [77]. Similarly, the essential oil of rosemary [*Rosmarinus officinalis* L. (Lamiaceae)] was poor active when incorporated into a nanoemulsion and tested for larvicidal activity. The mortality of L3 larvae at 250 µg/mL of product was 80% at 24 h and 90% at 48 h [76].

Seo et al. (2012) investigated the activity of 7 plant species essential oils against L3 larvae at 50 µg/mL and 100 µg/mL. Mortality ranged from 80 to 100% after 48 h. The species tested were *Ferula galbaniflua* Boiss. & Buhse and *Trachyspermum ammi* (L.) Sprague (Apiaceae); *Hyssopus officinalis* L. and *Pogostemon patchouli* Pellet. (Lamiaceae); *Larix europea* Lam. & A. DC. (Pinaceae); *Myroxylon pereirae* (Royle) Klotzsch (Fabaceae), and *Pelargonium graveolens* L’Hér. ex Aiton (Geraniaceae) [45]. Additional LC_50_ data is required to determine the degree of activity against *Ae. aegypti* larvae.

Repellent activity was reported for essentials oils of *Acantholippia seriphioides* (A. Gray) Moldenke (Verbenaceae) and *Aloysia citriodora* Palau (Verbenaceae); *Baccharis spartioides* (Hook. & Arn.) Remy (Asteraceae) and *Tagetes minuta* L. (Asteraceae), and *Minthostachys mollis* Griseb and *Rosmarinus officinalis*. All of the aforementioned demonstrated 100% repellency (12.5–50%) [63]. In another study, *Pluchea carolinensis* (Jack.) G. (Asteraceae) demonstrated repellency (36.6%) at 1.0% concentration and irritation (62.2%) at 0.1% concentration [79].

The results listed and discussed in this section clearly suggest that essential oils present a promising alternative to develop an effective natural and potentially more eco-friendly insecticide for the control of *Ae. aegypti*, especially during the larval phase. The challenges for these materials are to improve solubility in water and prolong the insecticidal effect. It is also important to understand the synergism and/or antagonism of their constituents, together with the optimum ratio.

## 5. Organic/Aqueous Extracts

Concerning organic/aqueous extracts, the plant families with the highest number of species tested against *Ae. aegypti* larvae were Fabaceae, Asteraceae, Piperaceae and Euphorbiaceae. Similarly, as described by Isman (2015), India was the country with the most publications in this field, followed by Brazil [14].

Of the 20 plant species, at least one organic/aqueous extract showed high larvicidal activity (LC_50_ < 50 µg/mL); 12 were active (LC_50_ 50–100 µg/mL) and 26 had weak activity (LC_50_ > 100 µg/mL). Nevertheless, these values can change significantly after formulation in a similar way to essential oils, as described in Section 8 “Limitations and/or Expectations of Plant Natural Product Insecticidal Applications”. 

Figure 4 details the chemical structures of the secondary metabolites identified in the organic extracts.

Table 3 and Table 4 summarise the scientific literature selected for the discussion of insecticidal activities of organic/aqueous extracts against *Ae. aeygpti* mosquito. Table 3 describes larvicidal activities, while Table 4 describes adulticidal, pupicidal, ovicidal, repellent and oviposition activities.

*Piper* species (Piperaceae) demonstrated LC_50_ ranging from 2.23 to 567 μg/mL for L3 and L4 larval stages [106,109,110]. The most active species extracts were *Piper longum* L. (fruit ethanolic), followed by *P. sarmentosum* (entire plant ethanolic LC_50_ 4.06 μg/mL) and *Piper ribesoides* Wall. (wood LC_50_ 8.13 μg/mL) [109]. *Piper nigrum* L. peppercorn ethanolic extract was active and purified fractions were highly active, with possible toxicity due to oleic acid (**18)** [110]. *P. aduncum* and *Piper hispidum* Sw. displayed weak activity against L3 larvae (LC_50_ > 150 μg/mL) [106].

An *Echinops transiliensis* Golosk. (Asteraceae) root dichloromethane extract showed strong toxicity against larvae (LC_50_ 3.21 μg/mL). The nine thiophene derivatives isolated showed strong activity (LC_50_ 0.16—19.97 μg/mL) [90]. Similarly, a *Euphorbia tirucalli* L. (Euphorbiaceae) stem bark petroleum ether extract presented LC_50_ 4.25 μg/mL against L4 larvae, while a *Euphorbia hirta* L. leaves extract exhibited weak activity (LC_50_ 272.36 μg/mL) [92].

The *Hypericum japonicum* Thunb. (Hypericaceae) species was also very toxic to mosquito larvae. The organic extracts of the whole plant presented LC_50_ values between 7.37 and 13.15 μg/mL. The methanolic extract proved the most active, its purified fraction displaying LC_50_ 0.95 μg/mL. The major constituents of this active fraction were isopropyl palmitate (**19**), 2,4-Di-*tert*-butylphenol (**20**) and hydrocinnamic acid (**21**) [97]. Similarly active, the *Nerine sarniensis* (L.) Herb. (Amaryllidaceae) bulb ethyl acetate extract demonstrated toxicity against L1 larvae (LC_50_ 8.0 μg/mL) and adult females (LC_50_ 4.6 μg/mosquito) [104].

Purified fractions of a *n*-hexane extract of *Limonia acidissima* L. (Rutaceae) leaves showed interesting ovicidal, larvicidal and pupicidal activities. The LC_50_ for L3 larvae ranged from 4.11 to 23.53 µg/mL; for pupae 4.19 to 39.48 µg/mL, and the maximum inhibition of egg hatching was 78.4% at 10 ppm. Bioguided fractionation resulted in the isolation of nilocetin (**88**) [99].

The biological activity of each plant species extract is specific to the plant part(s) and the polarity of the extraction solvent used. Furthermore, activity can differ significantly for the 4 different larval stages [58,86,89,92,93,94,96,98,100,108,113,115]. This variation is discussed below.

A *Jatropha curcas* L. (Euphorbiaceae) leaves petroleum ether extract showed LC_50_ 8.79 μg/mL whereas the root ethanolic extract demonstrated less activity (LC_50_ 44.75 μg/mL) [92,98]. A methanolic extract of *Gardenia ternifolia* Schumach. & Thonn. (Rubiaceae) leaves was very active (LC_50_ 32.01 μg/mL) whereas an acetone extract exhibited lower activity (LC_50_ 83.31 μg/mL) [94]. This study exemplifies the effect of different solvents on the toxicity of the same plant part.

*n*-Hexane, chloroform and acetone extracts of *A. nilotica* seed pods showed weak activity (LC_50_ 103—169 μg/mL), while ethyl acetate was more active (LC_50_ 59.12 μg/mL). Interestingly, a benzene extract demonstrated the most activity (LC_50_ 45.32 μg/mL) [58]. A *Ficus benghalensis* L. (Moraceae) leaves methanolic extract showed activity against L2, L3 and L4 larvae, but the activity was low for benzene and acetone extracts [93]. Another example of solvent variation was an aqueous extract of *Solanum nigrum* L. (Solanaceae) fruits which displayed very low activity (LC_50_ 359.0 μg/mL) against L3/L4 larvae, while higher activity was observed for a hexanic extract (LC_50_ 17.63 μg/mL) [113].

With not so discrepant, but nonetheless different results, a *Pemphis acidula J.R.* Forst. & G. Forst. (Lythraceae) leaves methanolic extract (LC_50_ 22.10 μg/mL) was more active than a benzenic leaves extract (LC_50_ 43.99 μg/mL) [108], while a *Tagetes patula* L. (Asteraceae) seeds acetone extract (LC_50_ 15.74 μg/mL) was more active than the 50% ethanol extract (LC_50_ 25.46 μg/mL) [93]. *P. acidula* extracts also demonstrated complete inhibition of egg hatchability at 500 ppm acetone and 450 ppm methanolic [108].

In addition, different parts of *Lonchocarpus urucu* Killip & A.C. Sm. (Fabaceae) extracted with the same solvent (methanol) showed different toxicity. The root bark extract was more active (LC_50_ 17.6 μg/mL) than the root medulla extract (LC_50_ 33.32 μg/mL) against L4 larvae [100]. The toxicity of *Heracleum rigens* Wall. (Apiaceae) seed extracts was evaluated against different larval stages (LC_50_ 40.64 to 308.65 μg/mL), with the petroleum ether extract the most toxic to all larval stages and acetone the least toxic [96].

Different organic solvent extracts of *Cassia fistula* L. (Fabaceae) leaves were evaluated against the mosquito (larvicide, ovicide and repellent). The methanolic extract was the most active for all activities, notably as a larvicidal (LC_50_ 10.69 μg/mL). Other extracts also demonstrated high activity against larvae: benzene (LC_50_ 18.27 μg/mL) and acetone (LC_50_ 23.95 μg/mL). The non-hatching concentration for eggs ranged from 120 to 160 mg/L and the repellent action (100% at 5 mg/cm^2^) ranged from 6.0 to 4.3 h [86].

*Dalbergia brasiliensis* Vogel (Fabaceae), commonly known as Jacarandá-da-Bahia in Brazil, is a tree native to the states of Bahia, Minas Gerais, Espírito Santo, Rio de Janeiro and São Paulo. Larvicidal activity of its leaves and trunk bark ethanolic extracts, together with fractions purified by partitioning with *n*-hexane, ethyl acetate and chloroform, were similar (LC_50_ between 24.0 and 44.0 μg/mL) [89].

Studies using the incorporation of inorganic nanoparticles, such as zinc oxide and silver in plant extracts, have shown an increase in their biological activity. They are generally easy to obtain, inexpensive, not to mention non-toxic to humans and animals [53]. All of the plant extracts described below showed higher larvicidal activity when incorporated into nanoparticles [53,82,116].

An aqueous extract of *Artemisia herba-alba* Asso (Asteraceae) leaves was tested against L4 larvae strains from India and Saudi Arabia. The LC_50_ values were 117.18 μg/mL and 614.54 μg/mL for India and Saudi Arabia larvae, respectively. When the extract was incorporated into silver nanoparticles the activity increased significantly to 10.70 μg/mL and 33.58 μg/mL, respectively. Similar results were observed against adult mosquitoes [82].

Aqueous extracts of *Ventilago madraspatana* Gaertn. (Rhamnaceae) and *Zeuxine gracilis* (Berda) Bl. (Orchidaceae) leaves were also more toxic to egg, larvae and adult mosquitoes when incorporated into silver nano particle. No egg hatching was observed at 120 μg/mL and 12 μg/mL, respectively. Corresponding larvicidal (LC_50_ 26.92 μg/mL and 10.39 μg/mL) and adulticidal effective dose activities (44.85 μg/mL and 27.90 μg/mL) were observed [112,116].

The activity of a zinc oxide nanoparticle incorporating a *Myristica fragans* leaf methanolic extract was compared with the crude extract. The activity of the crude extract against the 4 larvae stages (LC_50_ 162.03 to 273.9 μg/mL) was less than the nanoparticles (LC_50_ 3.44 to 10.28 μg/mL). Similar activity was reported against the pupa (crude extract LC_50_ 359.08 μg/mL and nanoparticles LC_50_ 14.63 μg/mL), and female adult forms (crude extract LC_50_ 180.26 μg/mosquito and nanoparticles LC_50_ 15 μg/mosquito) [53].

Following are the results of plant species that demonstrated at least one active extract for larvicidal action. A 90% ethanol extract of the leaves of *Scoparia dulcis* L. (Plantaginaceae), a plant used in Brazilian indigenous medicine, demonstrated activity against L4 larvae (LC_50_ 83.43 μg/mL). The compounds isolated were betulinic acid (**22**); scopadulcic acid A (**23**); scopadulcic acid B (**24**); scopadiol (**25**); scopadulciol (**26**) and scopadulin (**27**) [95].

A methanolic extract of *Cinnamosma fragrans* Baill. (Canellaceae) root bark demonstrated potential as an insecticide acting in different ways: L1 larvae (LC_50_ 52.5 μg/mL), adult (0.17 μg/mg) and 80% repellency at 20.8 μg/cm^2^ [87]. Similarly, extracts of *Ervatamia coronaria* (Jacq.) Stapf. (Apocynaceae), commonly known as Capre jasmine, were evaluated for different insecticidal applications. L3 larvicides: benzene (LC_50_ 89.59 µg/mL) and ethyl acetate (LC_50_ 97.53 µg/mL) [85]. Ovicide/no hatchability: methanol (200 ppm), benzene (250 ppm) and ethyl acetate (300 ppm) [85,117]. The repellent activity was the same for all extracts (100% repellency at 5 mg/cm^2^) [85,117].

Different extracts were obtained from *Mirabilis jalapa* L. (Nyctaginaceae) leaves and investigated for their L3 larvicidal activity: methanol (LC_50_ 64.58 µg/mL), ethyl acetate (LC_50_ 72.77 µg/mL), chloroform (LC_50_ 88.20 µg/mL) and benzene (LC_50_ 97.03 µg/mL) [103]. Extracts of three other species obtained from the same plant part with different organic solvents were also evaluated. The methanolic extracts were the most active: *Acalypha alnifolia* Klein ex Willd. (Euphorbiaceae) (L4 instar LC_50_ 128.55–202.15 µg/mL), *Orthosiphon thymiflorus* (Roth) Sleesen (Labiatae) (L3 instar LC_50_ 149.96–228.13 µg/mL) and *Ocimum sanctum* L. (Labiatae) (L4 instar LC_50_ 175.67–425.94 µg/mL) [80,105,107].

Petroleum ether leaf extracts of *Pedilanthus tithymaloides* (L.) Poit. (Euphorbiaceae), *Citrullus colocynthis* (L.) Schrad. (Cucurbitaceae) and *Phyllanthus amarus* Schumach. & Thonn. (Euphorbiaceae) were active against L4 instar, with LC_50_ 55.26 µg/mL, 74.57 µg/mL and 90.92 µg/mL, respectively [88,92]. Similar extracts of *Catharanthus roseus* (L.) G. Don (Apocynaceae), *Eupatorium odoratum* L. (Asteraceae), *Nyctanthes arbor-tristis* L. (Oleaceae), Boenninghausenia albiflora (Hook.) Rchb. ex Meisn. (Rutaceae) and *Valeriana hardwickii* Wall. (Valerianaceae) presented weak activity (LC_50_ > 100 µg/mL) [83].

Larvicidal activity was also reported for *Maytenus oblongata* Reissek (Celasteraceae) bark ethyl acetate extract (LC_50_ 74.4 µg/mL) and *Millettia pachycarpa* Benth. (Fabaceae) root ethanol extract (LC_50_ 98.47 µg/mL) [101,102]. The latter also demonstrated ovicidal activity at 200 ppm (100% non-hatchability) [102].

Ethanol, *n*-hexane and *n*-butanol extracts of flowers and aerial parts *of Buddleja polystachya* Fresen. (Buddlejaceae) were evaluated for adulticidal activity at 5 µg/mosquito: the most active was *n*-hexane aerial parts (100% mortality), followed by *n*-hexane flowers (96.7% mortality), ethanol aerial parts (90% mortality) and ethanol flowers (83.3% mortality). Only the n-butanol flower extract was investigated for L1 larvicidal activity, demonstrating 100% mortality at 1 µg/µL [84].

Finally, the species that presented poor larvicidal action for all tested extracts. *Aristolochia bracteata* Retz. (Aristolochiaceae) leaves methanolic extract was evaluated against L3 larvae (LC_50_ 114.89 µg/mL), egg (100% non-hatchability at 240 ppm) and adult mosquito (100% repellency at 6 mg/cm^2^) [81]. Similarly, *Caesalpinia pulcherrima* (L.) Sw. (Fabaceae) extracts were evaluated against larvae: benzene (LC_50_ 136.37 µg/mL) and ethyl acetate (LC_50_ 144.67 µg/mL), and 100% non-hatchability for benzene (375 ppm) and ethyl acetate (450 ppm). However, both extracts demonstrated 100% repellency at 5 mg/cm^2^ [85].

In another study involving a methanol extract of *C. pulcherrima*, complete inhibition of egg hatching was reported at 300 ppm. The repellency was the same as the aforementioned study (5 mg/cm^2^) [117]. *Coccinia indica* Wight & Arn. (Cucurbitaceae) presented similar insecticidal properties for different extracts, with a methanolic extract the most active in terms of ovicidal activity (zero hatchability at 200 ppm) and a hexanic extract having the more effective repellency (100% of repellency at 1 mg/cm^2^). For 100% non-hatchability, the concentrations were between 200 ppm and 300 ppm and for 100% of repellency were between 1 and 5 mg/cm^2^ [119].

The methanolic extract of *Eclipta alba* (L.) Hassk (Asteraceae) leaves was also the most active among the solvents of different polarities used to evaluate larvicidal and ovicidal activities of this plant. The LC_50_ values against L3 larvae were between 127 and 165 µg/mL. Complete inhibition of egg hatching occurred at 300 ppm for the methanolic extract and 350 ppm for the other solvents [91].

The methanolic extract of *Mentha piperita* L. (Lamiaceae) and different extracts of *Cardiospermum halicacabum* L. (Sapindaceae) showed repellent activity [118,120]. Essential oils and aqueous extracts of the red and pink flowers of *A. purpurata* were investigated for both larvicidal activity and oviposition effect. Similar to the essential oils, the extract of the pink flower was more active than the red, and both disrupted oviposition [67].

In general, organic extracts from different parts of *Parthenium hysterophorus* (Asteraceae), *Pithecellobium dulce* (Roxb.) Benth. (Fabaceae) and *Solanum xanthocarpum* Schrad. & J.C. Wendl. (Solanaceae) showed weak insecticidal action, requiring high concentrations to demonstrate some biological activity [111,114,121]. Other species that were inactive were *Helicteres velutina* K. Schum. (Malvaceae), *Momordica charantia* L. (Cucurbitaceae), *Ormosia arborea* Vell (Fabaceae), *Solanum variabile* Mart. (Solanaceae), *Spermacoce latifolia* Aubl. (Rubiaceae) and *Turnera ulmifolia* L. (Turneraceae) [88,95,106].

Several organic extracts have shown important insecticidal activities against the *Aedes* mosquito, including *E. transiliensis, E. tirucalli, H. japonicum, N. sarnisiensis, P. longum, P. ribesoides* and *P. sarmentosum*. Studies in this section show the potential of plant natural products as insecticides targeting different stages of the mosquito life cycle and how different formulation approaches, such as the incorporation of botanical extracts into silver and zinc nanoparticles, can increase the insecticidal effects. Prominent examples include nanoformulations of *Z. gracilis, M. fragans* and *A. herba-herba* extracts. These data reinforce the large diversity of plants with toxic effect in different life stages of the *Aedes aegypti* mosquito. However, an important consideration for these materials is the type of extraction solvent employed, such as n-hexane, chloroform, benzene, given their toxicity to humans associated with harmful residues [122].

## 6. Secondary Metabolites

### 6.1. Terpenes

Terpenoids are a very promising target for the development of products of natural origin to be used in the control of the *Ae. aegypti* mosquito. These compounds were the most identified in the essential oils, extracts and purified fractions, generally having better results against the mosquito, especially in terms of larvicidal activity. Of the terpenes, monoterpenes are the most active and present great possibilities in bioinsecticide applications due to their low toxicity against mammals and non-target organisms [50].

This significant activity against the mosquito can be explained by the hydrophobicity of this class. Terpene toxicity against *Ae. aegypti* larvae may be associated with their nonpolar property as reported for other insects [123,124]. This property increases the ability of the compound to penetrate the hydrophobic larvae cuticle and renders them more toxic to the insect in comparison to polar compounds [123]. The chemical structures of the terpenes tested are shown in Figure 5, Figure 6 and Figure 7.

Diterpene 7-oxo-8,11,13-cleistanthatrien-3-ol (**28**), isolated from the dichloromethane extract of *Vellozia gigantea* N.L. Menezes & Mello-Silva (Velloziaceae) adventitious roots, caused 100% larvae mortality at 416.06 µM [125]. The adulticidal activity of the diterpene phytol (**29**) was LC_50_ 4.23 µM/mosquito [84].

The triterpenoids ursolic acid (**30**) and betulinic acid (**22**) showed larvicidal activity with LC_50_ 245.24 µM and 310.83 µM, respectively. Their corresponding structures are illustrated in Figure 5. Bioassays of their chemical derivatives, with esterification of the hydroxyl group at the C-3 position, demonstrated less activity, suggesting that the hydroxyl group plays an important role in larvicidal activity [126].

The sesquiterpenes α-costic acid (**31**) and inuloxin A (**32**), both isolated from *Inula viscosa* (L.) Aiton (Asteraceae), demonstrated strong activity against L1 larvae. The concentration of each terpene required for 100% mortality was 4.27 µM and 4.03 µM, respectively [127]. Other sesquiterpene alkaloids with strong larvicidal activity (L3 and L4 instar) were 1-O-benzoyl-1-deacetyl-4-deoxyalatamine (**33**) (LC_50_ 9.4 µM) and 1,2-O-dibenzoyl-1,2-deacetyl-4-deoxyalatamine (**34**) (LC_50_ 2.3 µM). These sesquiterpenes with a β-dihydroagrofuran skeleton were isolated from *M. oblongata* stems [101].

β-caryophyllene (**8**) and caryophyllene oxide (**35**) demonstrated lower larvicidal activity, with LC_50_ values of 127.23 and 135.24 µM, respectively [78], together with the sesquiterpene cinnamodial (**36**), isolated from *C. fragrans* (LC_50_ 70 µM) [87]. Regarding adulticidal activity, sesquiterpenes isolated from *C. fragrans*—cinnamodial (**36**), cinnafragrin A (**37**) and cinnamosmolide (**38**)—showed strong activity with ED_50_ 0.29, 2.85 and 12.79 nmol/mg mosquito, respectively [87].

Monoterpenes were the most evaluated for larvicidal activity with LC_50_ values ranging from 88 to 540 µM. The terpene hydrocarbons: limonene (**13**) (LC_50_ 88.16 µM); α-terpinene (**39**) (LC_50_ 107.90 µM); α-phellandrene (**40**) (LC_50_ 121.85 µM), and ρ-cymene (**17**) (LC_50_ 143.05 µM) were the most active [61,72]. Terpinolene (**41**), γ-terpinene (**42**)**,** β-myrcene (**43**) and sabinene (**3**) showed LC_50_ 208.46 µM, 225.35 µM, 262.78 µM and 543.92 µM, respectively [61,71]. The oxygenated terpene hydrocarbons carvone (**11**) and cis-carveol (**12**) demonstrated activity with LC_50_ of 155.62 µM and 218.88 µM, respectively [72].

A study of the larvicidal activity of α-pinene and β-pinene enantiomers reported the following LC_50_ values: (-)-β-pinene (**44**) (263.52 µM) and (+)-β-pinene (**45**) (414.73 µM); (-)-α-pinene (**46**) (363.35 µM) and (+)-α-pinene (**47**) (484.26 µM). The (-) enantiomers displayed higher activity than (+), so these results showed that the type of enantiomer and even the racemic mixture could directly interfere with the activity [78].

The monoterpene limonene (**13**) was incorporated into a nanoemulsion to improve its water solubility and therefore increase its activity [68]. This compound deserves to be highlighted as it presented high oral LC_50_ (>4000 mg/kg) and dermatological LC_50_ (>5000 mg/kg) values for rodents. It is therefore considered safe and non-toxic to mammals [30].

### 6.2. Phenylpropanoids and Phenolic Derivatives

The chemical structures of the phenylpropanoids and esters discussed in this section are illustrated in Figure 8.

Among the phenylpropanoids and phenolic derivatives classes cinnamaldehyde (**48**) and cinnamyl acetate (**49**) can be highlighted as they presented interesting larvicidal activity (LC_50_ 219.43 and 187.27 µM, respectively) [64]. It is important to note that cinnamaldehyde is present in commercial insect-fighting formulations such as Cinamite^®^ and Valero^®^. This information suggests that these secondary metabolites have a high potential for use against *Ae. aegypti* due to their possible toxicological safety, given that they have been authorized as insecticides since 2001.

Important larvicidal properties were also reported for eugenol (**15**), a phenolic compound that presents some advantages such as its non-persistence in water and soil, together with its natural degradation in organic acids through the action of *Pseudomonas,* a soil-dwelling bacterium. Furthermore, it is 1500 times less toxic than pyrethrins and 15,000 times less toxic than azinphos-methyl, an organophosphate [50]. The LC_50_ value for larvicidal activity was 200.97 μM [64]. The phenylpropanoid *trans*-anethole (**7**) also showed important action against *Ae. aegypti* larvae (LC_50_ 283.40 μM) [64].

Metabolites isolated from *Zingiber officinale* Roscoe (Zingiberaceae) demonstrated strong larvicidal activity, presenting LC_50_ values of 15.96 µM for 4-gingerol (**50**), 37.36 µM for 6-dehydrogingerdione (**51**) and 61.86 µM for 6-gingerol (**52**) [128]. The phenolic derivatives esters benzyl benzoate (**53**) and benzyl cinnamate (**54**)—demonstrated interesting activity as their 100% larvae mortality concentrations were 117.79 μM and 104.92 μM, respectively [45].

### 6.3. Alkaloids and Amides

The Piperaceae family has numerous compounds with promising activity against *Ae. aegypti*. The chemical structures of the aforementioned alkaloids and amides are illustrated in Figure 9.

N-Isobutylamide alkaloids from *Piper* species presented potent larvicidal and adulticidal activities. Secondary metabolites isolated from *P. nigrum*: pellitorine (**55**), guineensine (**56**), pipercide (**57**) and retrofractamide A (**58**) presented respective larvicidal LC_50_ values of 4.12 μM, 2.32 μM, 0.28 μM and 0.12 μM [129]. Regarding the adulticidal activity, LC_50_ values (μM/female mosquito) were 0.76, 4.43, 6.11 and 4.22, respectively [130]. Pipernonaline (**59**) isolated from the methanolic extract of *P. longum*, also showed potent larvicidal activity (LC_50_ 0.73 μM) [127], while piperine (**60**) and pipwaqarine (**61**) isolated from *P. nigrum* demonstrated LC_50_ of 17.87 and 75.46 μM, respectively [131,132].

The analysis of the structure-activity relationship for the N-isobutylamide alkaloids **55**–**61**, it is reasonable to hypothesise that the N-isobutylamine moiety is of crucial importance in terms of larvicidal activity, while the methylenedioxyphenyl moiety does not appear to be essential.

The mesembrine-type alkaloid sarniensinol (**62**) isolated from *N. sarniensis* exhibited strong larvicidal (LC_50_ 24.24 μM) and adulticidal (LC_50_ 13.88 μM/female mosquito) activities [104]. The crinine-type alkaloid crinsarnine (**63**) only demonstrated strong adulticidal activity (LC_50_ 5.78 μM/female mosquito) [133]. N-hydroxyaristolactam I (**64**), an aristololactam derivative, also showed strong larvicidal activity (LC_50_ 11.45 μM) [130], whereas the pyrrolidine alkaloid (*Z*)-3-(4-hydroxybenzylidene)-4-(4-hydroxyphenyl)-1-methylpyrrolidin-2-one (**65**) demonstrated weak activity (LC_50_ 785.86 μM) [134,135].

### 6.4. Thiophenes and Acids

Thiophene and fatty acid chemical structures are illustrated in Figure 10.

Nine thiophenes, with different numbers of thiophene rings, isolated from *E. transiliensis* exhibited strong larvicidal activity and a positive correlation was reported between the number of thiophene rings and larvicidal activity, with thiophene derivatives composed of more rings demonstrating more activity [90].

The terthiophene 2,2′: 5′,2′′-terthiophene (**66**) was the most active (LC_50_ 0.65 μM). The activity (LC_50_) of bithiophenes was: 4-(2,2′-bithiophen-5-yl)but-3-yne-1,2-diyl diacetate (**67**) (12.54 μM); 4-(2,2′-bithiophen-5-yl)-2-hydroxybut-3-yn-1-yl acetate (**68**) (25.31 μM) and 4-(2,2′-bithiophen-5-yl)but-3-yne-1,2-diol (**69**) (39.19 μM). Lower larvicidal activity was observed for monothiophenes: 2-chloro-4-[5-(penta-1,3-diyn-1-yl)thiophen-2-yl]but-3-yn-1-yl acetate (**70**) (49.56 μM); 4-[5-(penta-1,3-diyn-1-yl)thiophen-2-yl]but-3-yne-1,2-diyl diacetate (**71**) (56.02 μM); 4-[5-(penta-1,3-diyn-1-yl)thiophen-2-yl]but-3-yne-1,2-diol (**72**) (56.68 μM); 2-hydroxy-4-[5-(penta-1,3-diyn-1-yl)thiophen-2-yl]but-3-yn-1-yl acetate (**73**) (66.64 μM) and 1-hydroxy-4-[5-(penta-1,3-diyn-1-yl)thiophen-2-yl]but-3-yn-2-yl acetate (**74**) (71.74 μM) [90].

Tetradecanoic acid (**75**) showed both larvicidal action (LC_50_ 131.37 μM) and oviposition attraction (78.2% at 43.79 μM) [136,137]. Other fatty acids, hexadecanoic acid (**76**) and dodecanoic acid (**77**) displayed lower attraction: 57.4% for (**76**) at 3.9 μM and 68.8% for (**77**) at 249.6 μM [137].

### 6.5. Flavonoids

The corresponding chemical structures are shown in Figure 11. Quercetin-4′,7-O-dimethyl ether (**78**), naringenin-7-O-methyl ether (**79**) and kaempferol-7-O-methyl ether (**80**) flavonoids isolated from *G. ternifolia* were active against L2 larvae. The LC_50_ values were 108.09 μM, 85.37 μM and 102.08 μM, respectively [94].

### 6.6. Neolignans

Figure 12 details the neolignan chemical structures.

Eupomatenoid-6 (**81**), a neolignan isolated from *Piper solmsianum* C. DC. (Piperaceae), demonstrated strong larvicidal acitivity (LC_50_ 19.33 μM) with probably low toxicity to mammals (IC_50_ 39.30 μM for human fibroblast cells, MRC5, with an estimated LD_50_ of 42.26 mmol/kg) [138]. Grandisin (**82**) presented larvicidal activity (LC_50_ 346.82 μM). Histological analysis revealed that this neolignan damages the anterior-middle midgut of the larvae [139].

### 6.7. Furanochromones and Furanocoumarin

The chemical structures of the coumarins are illustrated in Figure 13.

Khellin (**83**), a natural furanochromone isolated from *Ammi visnaga* (L.) Lam. (Umbelliferae), caused L3 mortality (LC_50_ 192.1 μM) [140]. In another study, **83** demonstrated 100% larvae mortality at 3.84 μM and 75% adult mortality at 19.21 μM/mosquito for permethrin-susceptible strains [141]. Another furanochromone isolated from *A. visnaga*, visnagin (**84**) presented 93% mortality at 4.34 μM, together with moderate adulticidal activity (65% mortality at 21.72 μM/mosquito). In addition, this study investigated 2 furanocoumarins isolated from *Ruta graveolens* L. Royle (Rutaceae): 5-methoxypsoralen (5-MOP) (**85**) and 8-methoxypsoralen (8-MOP) (**86**), which showed moderate adulticidal activity (55% mortality at 23.13 μM/mosquito and 67.5% mortality at 23.13 μM/mosquito, respectively). Compound **86** presented weak activity against L3 (53.3% mortality at 4.63 μM) while **85** was inactive [141].

### 6.8. Other Secondary Metabolites

The chemical structures of the compounds referenced in this section are illustrated in Figure 14.

Several other classes of plant natural products have also been investigated regarding their insecticidal activities. Naphthoquinone 2-methoxy-1,4-naphthoquinone (**87**) isolated from *Impatiens glandulifera* (Balsaminaceae) showed extremely potent larvicidal acitivity (LC_50_ 0.45 μM) and moderate adulticidal activity (40% mortality at 26.6 μM) [127].

Nilocetin (**88**), a protolimonoid isolated from *L. acidissima*, also demonstrated very strong larvicidal (LC_50_ 0.96 μM) and pupicidal (LC_50_ 1.36 μM) activity which was higher than temephos, a well-documented chemical insecticide. The triterpenoid also caused 83% egg mortality at 4.38 μM [99]. Already a study evaluating volatile plant metabolites capable of eliciting an *Ae. aegypti* behavioral response reported that acetophenone (**89**) attracted adult mosquitoes whereas 1-octanol (**90**) acted as a repellent (flight aversive response) [142].

## 7. Mechanisms of Action

The mechanisms of action for the *Ae. aegypti* control relate more to the use of conventional chemical insecticides. Table 5 summarizes the mechanisms of action data of the botanical samples discussed in this section. *Ae. aegypti* control relies primarily on the use of conventional chemical insecticides which target different critical sites in the mosquito life cycle. Organophosphates and carbamates, for example, target acetylcholinesterase enzyme inhibition. Pyrethroids and some organochlorines target sodium channels. Cyclodienes and polychloroterpenes target gamma-aminobutyric acid (GABA) receptors [34].

Other mechanisms of alternative insecticides authorized by regulatory agencies vary in terms of their action. For example, a biological approach employs the use of entomopathogenic bacteria (*Bacillus thuringiensis israelensis* and *Bacillus sphaericus*), which act via the toxic action of their spores damaging the intestinal epithelium of larvae. Insect growth regulators (IGR) differ in that they inhibit insect chitin synthesis, and therefore disrupt the moulting process, while juvenile hormone analogs (JHA) act by interfering with the insect’s endocrine system [138].

Further studies are required in order to completely understand the various toxic action mechanisms of botanical insecticides. However, a number of mechanisms have been proposed and proven. The majority of mechanism of action studies have focused on the larval stage, particularly feeding and/or contact. In the case of ingestion, the action is usually through digestive toxicity whereas contact may involve enzymatic inhibition, endocrine disruption (acting especially during the moulting process), toxicity to the nervous system and other mechanisms depending on the target site [143].

The rapid toxic action of essential oils against the insect indicates a possible neurotoxic mode of action [144]. Phytochemicals may act in cholinergic, GABA, mitochondrial and octopaminergic systems [145]. A study of five volatile compounds commonly found in plant essential oils—eugenol, geraniol, coumarin, eucalyptol and carvacrol—investigated docking against octopamine and acetylcholinesterase receptors in *Ae. aegypti* and *Homo sapiens* protein models. All compounds were found to dock in both protein models, with some more selectivity for insect proteins [146].

Effects on the larval nervous system were observed after treatment with *Piper* species extracts. Tremor, convulsion, excitement, followed by paralysis and death were verified after exposure of larvae to *P. longum*, *P*. ribesoides and *P. sarmentosum* extracts. In addition, the larvae showed morphological changes in the anal papillae [109].

Essential oils of *I. verum, P. dioica* and *M. fragrans* inhibited acetylcholinesterase causing acetylcholine accumulation in the synapses, with the membrane in a constant state of excitement, culminating in ataxia, lack of neuromuscular coordination and eventual death [54]. The neurotoxic effect was also observed for a nanoemulsion with *P. emarginatus* essential oil. It probably causes reversible inhibition of acetylcholinesterase and consequently larval death [66].

*D. brasiliensis* extracts caused external morphological alterations in the larvae, resulting in interference in the moulting process. The authors also reported digestive toxicity and morphological changes in the anal papillae and respiratory siphon of the larvae which interfered with swimming and oxygen flow [89]. Similarly, nilocetin (**88**) (Figure 14) induced morphological deformations together with moulting symptoms and growth disruption in all mosquito life cycle stages. These compounds also totally ruptured the peritrophic membrane [99].

A *Lonchocarpus urucu* extract caused disruption in the peritrophic matrix, a medium intestine lining composed of chitin and proteins, whose functions are to protect against abrasion caused by food and micro-organisms, among others such as decreasing the excretion of digestive enzymes through their recycling. In addition, this extract caused extensive damage to the midgut epithelium (Table 5) [100].

**Table 5 molecules-25-03484-t005:** Mechanisms of action of botanicals against the *Ae. aegypti* mosquito.

Target Site	Mechanism of Action	Compound	Plant Species	Reference
Nervous system	Inhibition of acetylcholinesterase (AChE)	Essential oil	*Illicium verum*, *Pimenta dioica* and *Myristica fragrans*	[54]
		Nanoemulsion with essential oil	*Pterodon emarginatus*	[66]
	Not specified	Ethanolic extract	*Piper longum*, *Piper ribesoides* and *Piper sarmentosum*	[109]
Gut trypsin	Inhibition of trypsin and consequent decreased absorption of nutrientes and essential aminoacids	Aqueous extract	*Moringa oleifera*	[147]
		Essential oil	*Croton rhamnifolioides*	[70]
Peritrophic matrix	Change in internal morphology and consequent insect protection dysfuntion	Methanol extracts	*Derris (Lonchocarpus) urucu*	[100]
		Nilocetin	*Limonia acidissima*	[99]
Midgut epithelium	Tissue destruction and cell disorganization	Methanol extracts	*Derris (Lonchocarpus) urucu*	[100]
		Pellitorine	*Asarum heterotropoides*	[148]
		Grandisin	*Piper solmsianum*	[139]
Anal papillae	Morphological changes, interference with the larva swin	Crude extracts	*Dalbergia brasiliensis*	[89]
		Ethanolic extract	*Piper longum*, *Piper ribesoides* and *Piper sarmentosum*	[109]
Respiratory siphon	Morphological changes, interference with the oxygen flow	Crude extracts	*Dalbergia brasiliensis*	[89]
		Nanoemulsion with limonene	*Baccharis reticularia*	[68]
Anal gills	Comprehensive damage; debris in hemolymph	Pellitorine	*Asarum heterotropoides*	[148]
Thorax and exoskeleton	Changes in external morphology, interfering with the molting process	Crude extracts	*Dalbergia brasiliensis*	[89]
		Pellitorine	*Asarum heterotropoides*	[148]
		Nilocetin	*Limonia acidissima*	[99]
		Nanoemulsion with limonene	*Baccharis reticularia*	[68]
Digestive system	Digestive toxicity	Crude extracts	*Dalbergia brasiliensis*	[89]

Pellitorine (**55**), an isobutylamide alkaloid, whose structure is illustrated in Figure 9, promoted histological changes in the thorax, midgut and anal gills. These toxic effects probably occur as a result of compound action on the larval osmoregulation system [148]. Already a nanoemulsion with limonene (**13**) (Figure 7) promoted morphological alterations to the head, siphon, abdomen cuticles and thorax, promoting larvae fragility and low mobility [68].

*C. rhamnifolioides* essential oil induced toxicity in the larvae by trypsin-like activity. Trypsin is a serine protease that widely occurs in the gut of insects. A decrease in its activity may result in poor nutrient absorption and non-availability of essential aminoacids, causing insect death [72]. A *Moringa oleifera* (Moringaceae) extract also caused larval toxicity by inhibiting trypsin in the gut [147]. After treatment with grandisin (**82**) (Figure 12), larvae presented intense tissue destruction and cell disorganization in the anterior midgut [139].

## 8. Limitations and/or Expectations of Plant Natural Product Insecticide Applications

As demonstrated in this review, natural products of botanical origin are promising for control of the *Ae. aegypti* mosquito, although there remain several limitations and challenges to overcome for their application as insecticidal products. From 1998 until early 2011, the number of patents of essential oil-containing mosquito repellent inventions has almost doubled every 4 years [149], but the number of new products does not reflect this. There are several possible reasons for this disparity, such as: (i) the onerous regulatory processes involved in the registration of a pesticide product; (ii) the quantity of raw material biomass required to obtain sufficient extract and/or its isolated active compound, and (iii) most of the research is conducted at the laboratory scale often without field evaluation to confirm the product application [13,47,150].

The complex process of registering an insecticide discourages companies from investing in new products, especially in some places such as Brazil and the European Union. There are numerous criteria, including provision of non-target toxicology and environmental destination data, extensive data to guarantee plant stability and extract standardization, together with physico-chemical and microbiological procedures establishing quality control of the raw material and final product [47,48].

The low availability of raw materials due to limited yields and cultivation usually makes botanical insecticides more expensive than chemicals. The study of bioactive compound synthesis through biotechnology, such as tissue culture in bioreactors, constitutes an alternative to this limitation [150,151]. In moving from laboratory to industrial scale, a number of different factors must be considered: botanical material, analysis technique, formulations, toxicological tests, mechanisms of action, among others.

Considering the botanical material, it is essential to correctly identify the botanical species and determine the chemical composition of the extract (standardization) [13,50]. Chemical composition may vary depending on numerous factors, such as crop period, seasonality, phenological stage, temperature, humidity, luminosity, altitude, pluviometry, ultraviolet radiation, soil and nutrient conditions, geographical locations, collection method, drying and the part of the plant used, among others, and consequently impact insecticidal activity [152,153,154,155].

Regarding larvicidal analysis techniques, important considerations are: (i) larval phase, (ii) analysis time, and (iii) the use of positive and negative controls. In different studies, the younger the larval stage, the more susceptible it is to toxic effects, as reported for *Ficus benghalensis*, *Heracleum rigens*, *Myristica fragans* and *Solanum xanthocarpum* (Table 3) [53,93,96,114]. This characteristic may relate to the reduced feeding of larvae in the late L4 instar. If the toxic effect of the insecticide is ingestion-dependent, the effect may be less pronounced the closer the larva is to the pupa stage of metamorphosis [156]. Although most studies use 24 h as the contact time to express the mortality result, it is important to note that some materials may have delayed activity, actually causing larvae death after 48, 72 or even 96 h. Thus, during product development, it is necessary to assess the toxic effect at different time intervals. Finally, the use of negative and positive controls is essential to ensure results reliability, although a number of studies did not report this data [157]. Therefore, test non-uniformity makes it difficult to compare the results of different studies. This constitutes another obstacle to overcome for the development of plant natural product insecticides [158].

Understanding the mechanism of action is fundamental in using a material as an inseticidal product. Knowledge of the mechanism of action makes it possible to understand which non-target organisms could be harmed by the use of such products [145,159]. In addition, this information facilitates prospecting other possibly more active materials using biotechnology tools and in silico models [160,161,162]. However, in general it is not easy to understand the mechanisms of action of plant natural products. Normally there are multiple modes of action pertaining to the complex composition of the materials [30,49,145], that usually occur in different target sites, as described for *Piper* spp, *Derris (Lonchocarpus) urucu*, *Asarum heterotropoides* and *Dalbergia brasiliensis* (Table 5) [89,100,109,148].

During the development stage, it is essential to evaluate toxicity in non-target organisms for promising insecticides using suitable models, such as the fish embryo acute toxicity (FET) test [163]. This model has been proposed to determine the acute or lethal toxicity of materials in the embryonic stages of zebrafish (*Danio rerio*) and for environmental assessments [163,164]. In addition, it is important to consider other aquatic and terrestrial organisms according to the intended application location, such as fish, amphibians, bees, birds and mammals [159]. Considering that natural product insecticides have natural degradation mechanisms, they possibly present advantages in comparison with insecticides of synthetic origin [26,30,50].

In general, raw materials (essential oils, extracts and isolated compounds) from plant natural products are poorly soluble in water and do not persist in the environment, which complicates the application and reduces the effectiveness of the desired action [26,51,165]. Therefore, the use of pharmaceutical technology is of fundamental importance in the development of formulations. Among the techniques used, nanotechnology, encapsulation and use of hydrophobic matrices with an extended and controlled release system should be highlighted as they can prolong the residual effect of formulations [51,149,165,166,167] due to controlled release. Formulation development of natural products also poses a challenge for the application of these materials but it is imperative to improve both efficiency and cost-effectiveness [150,165]. Investing in botanical natural product formulations is an important advance in increasing the availability of commercial eco-friendly insecticides for *Ae. aegypti* control.

## 9. Conclusions

Considering the several stages of the insect development, the larvicidal test is the most evaluated bioassay in the search for insecticides to control *Ae. aegypti* for a number of reasons: (i) the larval phase is the longest in the immature stage; (ii) larvae are generally more sensitive to the toxic effects of compounds, and (iii) larvae breeding sites are localized and usually accessible. The search for ovicidal action is complex, especially due to its composition that hinders the toxic action of compounds. For the adult phase, there are compounds that cause toxicity by contact as well as those with repellent action.

Some very common edible botanical species such as *Petroselinum crispum*, *Foeniculum vulgare*, *Curcuma longa*, *Mentha spicata*, *Ocimum gratissimium* and *Rosmarinus officinalis* are highlighted, especially in the larval phase of *Ae. Aegypti*, due to their possible low toxicity to non-target organisms. However, other non-edible species have shown strong larvicidal extract activity, among them *Echinops transiliensis*, *Piper* ssp, *Hypericum japonicum* and *Nerine sarniensis.*

Essential oils provide a promising source for insecticidal applications due to their important insecticidal activities and possible toxicological safety for mammals and the environment. Moreover, they generally possess high oral and dermal LC_50_ values for these animals and are more readily degraded by natural ecosystem mechanisms.

Among the secondary metabolites, terpenes, especially monoterpenes, and phenylpropanoids are highlighted for larvicidal activity. These compounds are present in large quantities in essential oils. In addition, thiophenes, amides and alkaloids demonstrate high larvicidal and adulticidal activity.

Regarding the mechanisms of action, botanical natural products extracts, and pure compounds have displayed acitivities that include altering insect morphogenesis and therefore impairing the moulting process, respiration, feeding, and self-defense, among others. In addition, they altered biochemical processes and the nervous system.

Despite the limitations and obstacles to overcome, plant natural products are a suitable alternative source of eco-friendly botanical insecticides to control the *Ae. aegypti* mosquito, popularly known as dengue mosquito. Ever increasing mosquito resistance to conventional chemical insecticides warrants alternative products, which are safer for the environment and pose less risk to human health.

## Figures and Tables

**Figure 1 molecules-25-03484-f001:**
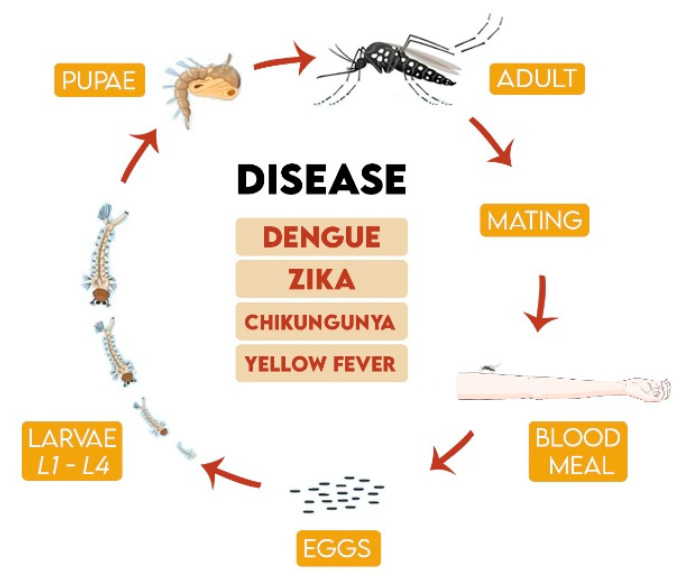
*Aedes aegypti* life cycle and the main arboviruses transmited by the female mosquito.

**Figure 2 molecules-25-03484-f002:**
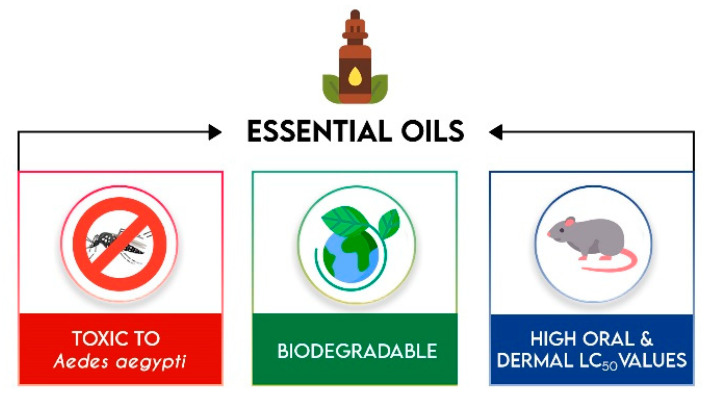
Essential oils to combat the *Ae. aegypti* mosquito: an ecologically safe alternative.

**Figure 3 molecules-25-03484-f003:**
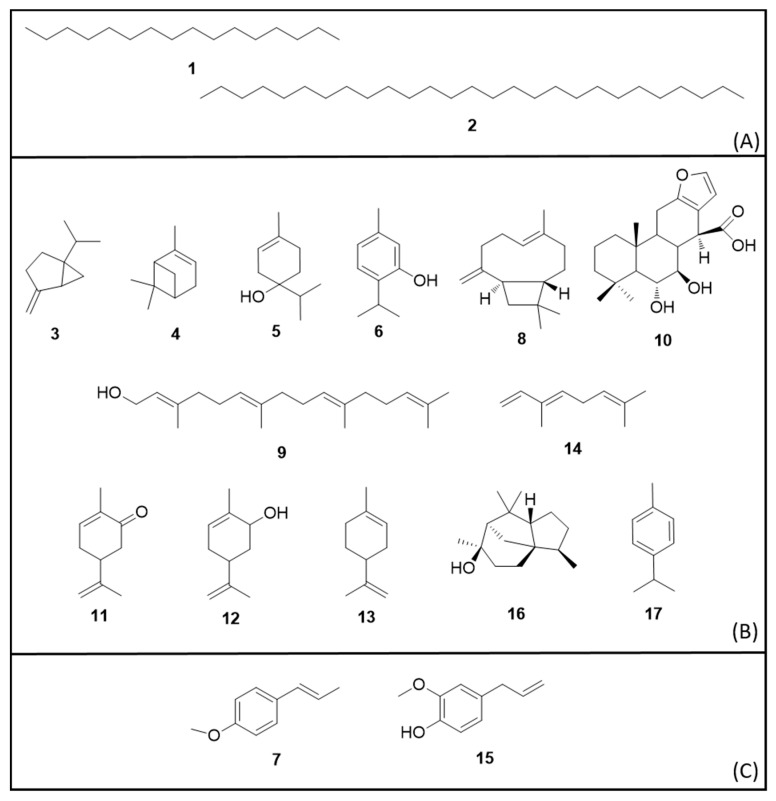
Secondary metabolites identified in essential oils with insecticidal activity against the *Ae. aegypti*. (**A**) Alkanes (**B**) Terpenes and (**C**) Phenylpropanoids.

**Figure 4 molecules-25-03484-f004:**
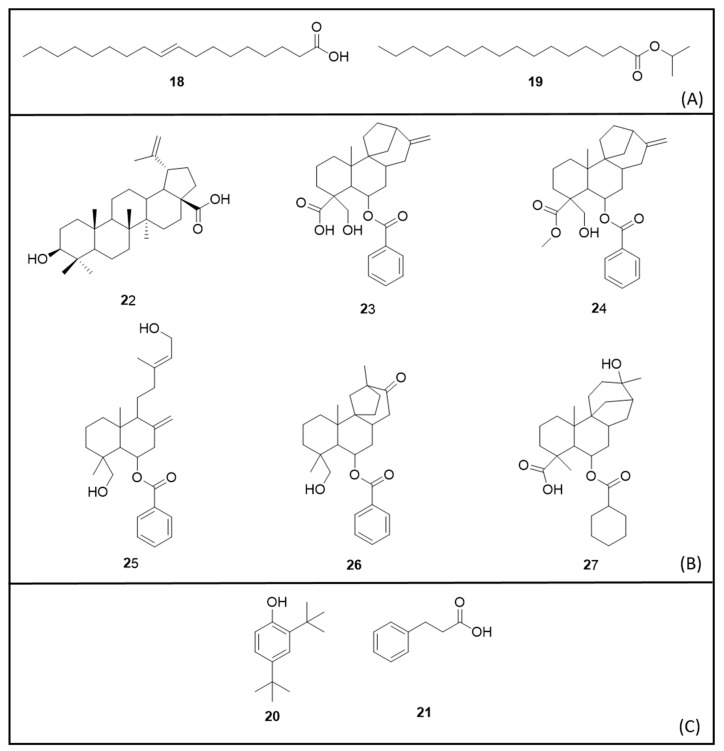
Secondary metabolites identified in organic extracts with insecticidal activity against *Ae. aegypti*. (**A**) Fatty acid and derivatives, (**B**) Diterpenes and triterpenes and (**C**) Others.

**Figure 5 molecules-25-03484-f005:**
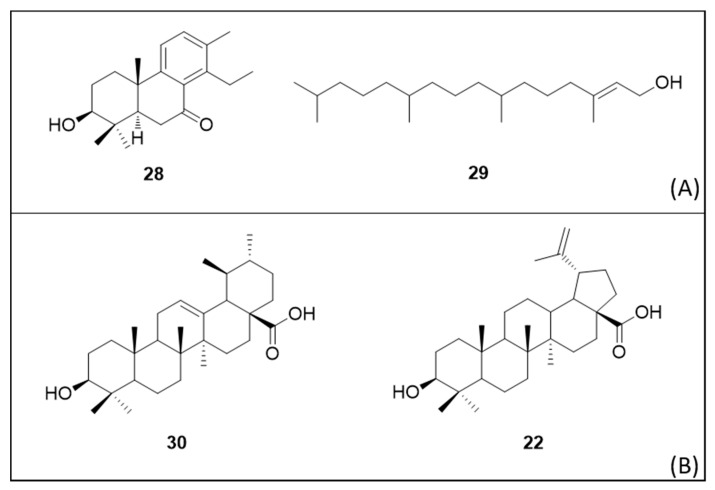
Diterpenes (**A**) and triterpenes (**B**) active against *Ae. aegypti*.

**Figure 6 molecules-25-03484-f006:**
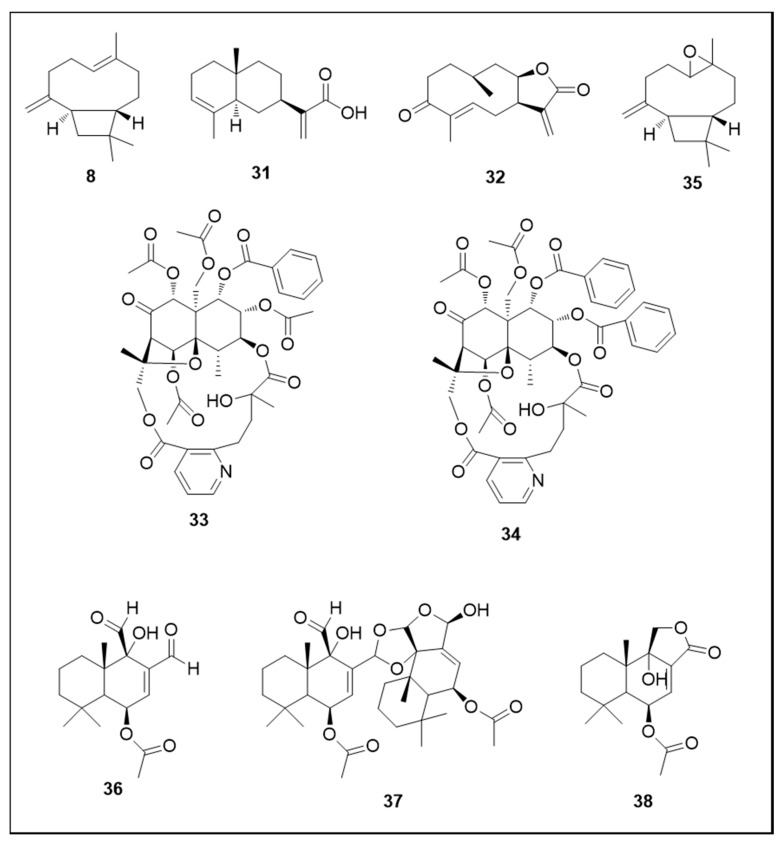
Sesquiterpenes active against *Ae. aegypti*.

**Figure 7 molecules-25-03484-f007:**
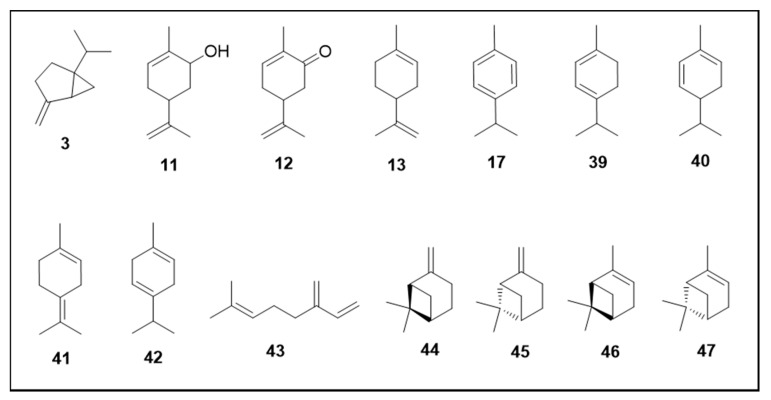
Monoterpenes active against *Ae. aegypti*.

**Figure 8 molecules-25-03484-f008:**
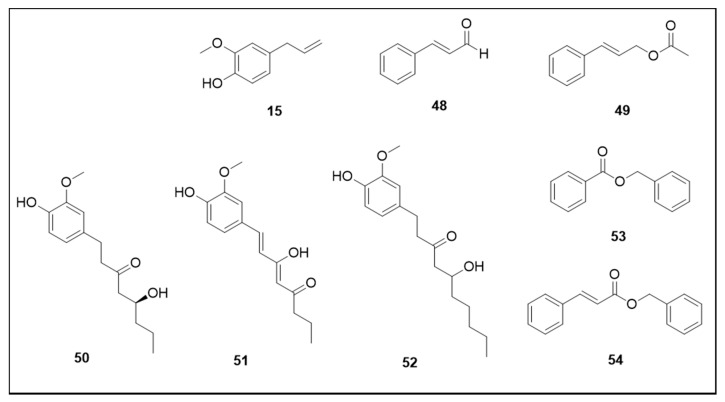
Phenylpropanoids and phenolic derivatives active against *Ae. aegypti*.

**Figure 9 molecules-25-03484-f009:**
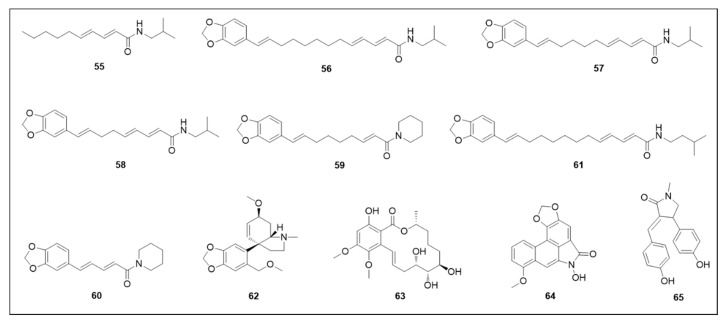
Alkaloids and amides active against *Ae. aegypti*.

**Figure 10 molecules-25-03484-f010:**
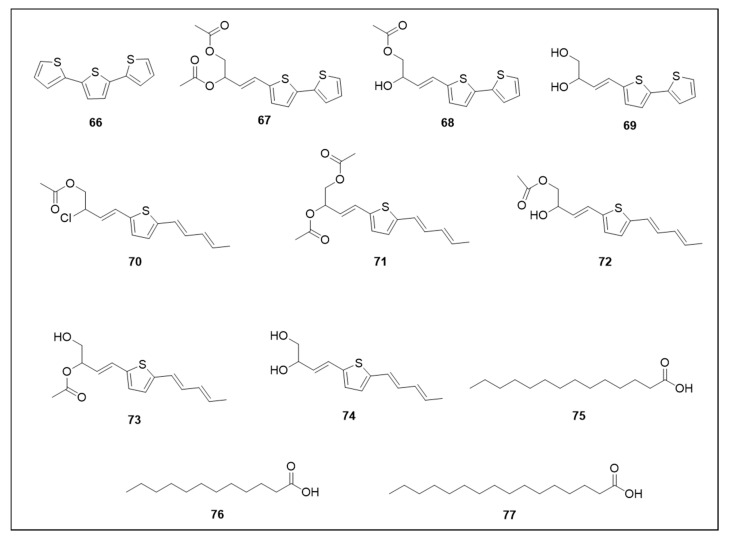
Fatty acids and thiophene derivatives active against *Ae. aegypti*.

**Figure 11 molecules-25-03484-f011:**
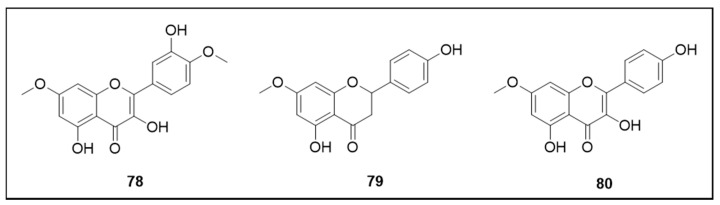
Flavonoids active against *Ae. aegypti*.

**Figure 12 molecules-25-03484-f012:**
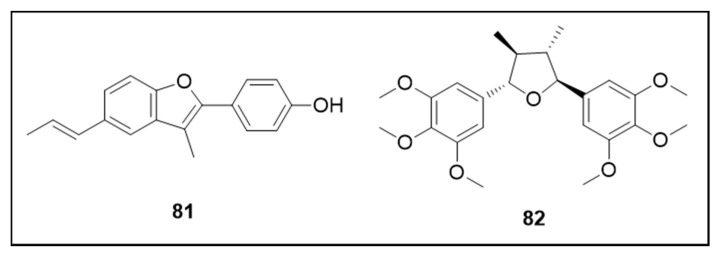
Neolignans active against *Ae. aegypti*.

**Figure 13 molecules-25-03484-f013:**
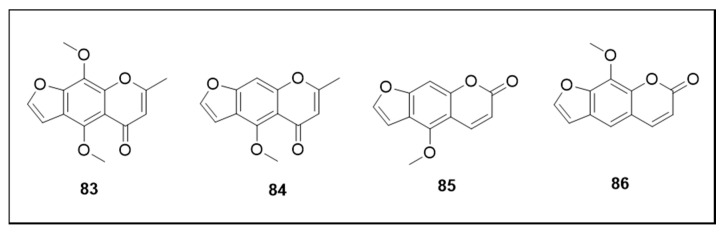
Furanochromones and furanocoumarins active against *Ae. aegypti*.

**Figure 14 molecules-25-03484-f014:**
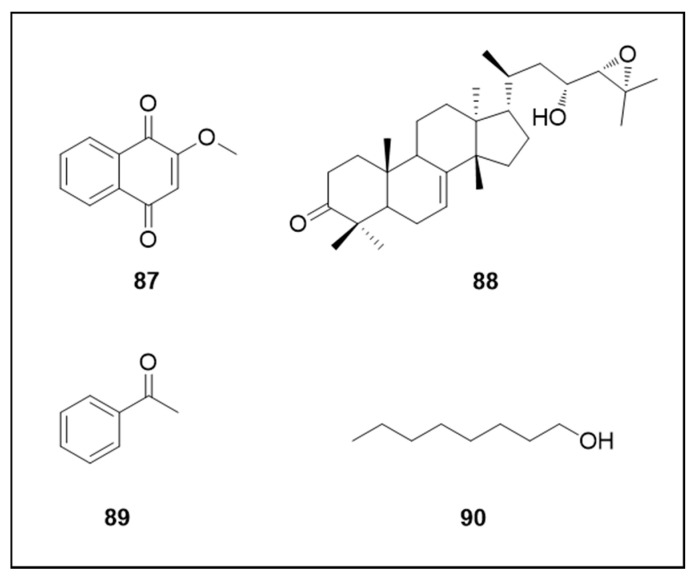
Naphthoquinone, protolimonoid, alcohol and ketone with activity against *Ae. aegypti*.

**Table 3 molecules-25-03484-t003:** Larvicidal activity of organic/aqueous extracts against the *Ae. aegypti* mosquito.

Plant Species	Family	Country	Part Used	Extraction Solvent	Larval Stage	Mortality		Time (h)	Reference
						LC_50_ (ppm)	LC_90_ (ppm)		
*Acacia nilótica* (L.) Delile	Fabaceae	India	Seed pod	*n*-Hexane	L4	169.25	201.62	24	[58]
				Benzene	L4	45.32	99.32	24	[58]
				Chloroform	L4	158.13	198.24	24	[58]
				Ethyl acetate	L4	59.12	75.82	24	[58]
				Acetone	L4	103.68	162.03	24	[58]
*Acalypha alnifolia* Klein ex Willd.	Euphorbiaceae	India	Leaves	*n*-Hexane	L4	202.15	476.57	24	[80]
				Chloroform	L4	182.58	460.83	24	[80]
				Ethyl acetate	L4	160.35	440.78	24	[80]
				Acetone	L4	146.07	415.38	24	[80]
				Methanol	L4	128.55	381.67	24	[80]
*Aristolochia bracteata* Retz.	Aristolochiaceae	India	Leaves	Methanol	L3	114.89	216.24	24	[81]
*Artemisia herba-alba* Asso	Asteraceae	Saudi Arabia	Leaves	Water	L4—India	117.18	227.63	24	[82]
				Water	L4—Saudi Arabia	614.52	1273.33	24	[82]
				Water + AgNP	L4—India	10.70	21.24	24	[82]
				Water + AgNP	L4—Saudi Arabia	33.58	57.0	24	[82]
*Boenninghausenia albiflora (Hook.)* Rchb. ex Meisn.	Rutaceae	India	Leaves	Petroleum ether	L4	125.0	190.0	ND	[83]
*Buddleja polystachya* Fresen.	Buddlejaceae	Saudi Arabia	Flowers	*n*-Butanol	L1	ND	ND	ND	[84]
*Caesalpinia pulcherrima* (L.) Sw.	Fabaceae	India	Leaves	Ethyl acetate	L3	144.67	276.99	24	[85]
				Benzene	L3	136.36	272.15	24	[85]
*Cassia fistula* L.	Fabaceae	India	Leaves	Methanol	L3	10.69	20.47	24	[86]
				Benzene	L3	18.27	35.67	24	[86]
				Acetone	L3	23.95	47.13	24	[86]
*Catharanthus roseus* (L.) G. Don	Apocynaceae	India	Leaves	Petroleum ether	L4	145.0	255.0	24	[83]
*Cinnamosma fragrans* Baill.	Canellaceae	Madagascar	Root Bark	Methanol	L1	52.5	ND	24	[87]
*Citrullus colocynthis* (L.) Schrad.	Cucurbitaceae	India	Leaves	Petroleum ether	L4	74.57	538.30	24	[88]
*Cunninghamia konishii* Hayata	Taxodiaceae	Taiwan	Wood	Ethanol	L4	240.0	>400.0	24	[73]
			Leaves	Ethanol	L4	>400.0	>400.0	24	[73]
*Dalbergia brasiliensis* Vogel	Fabaceae	Brazil	Leaves	Ethanol	L3	30.0	91.0	24	[89]
				*n*-Hexane Fraction	L3	44.0	81.0	24	[89]
				Chloroform Fraction	L3	33.0	75.0	24	[89]
				Ethyl acetate Fraction	L3	24.0	66.0	24	[89]
			Bark	Ethanol	L3	32.0	71.0	24	[89]
				*n*-Hexane Fraction	L3	31.0	72.0	24	[89]
				Chloroform Fraction	L3	25.0	50.0	24	[89]
				Ethyl acetate Fraction	L3	28.0	93.0	24	[89]
*Echinops transiliensis* Golosk.	Asteraceae	Kazakhstan	Root	Dichlorometane	ND	3.21	6.81	24	[90]
*Eclipta alba* (L.) Hassk	Asteraceae	India	Leaves	Benzene	L3	151.38	274.34	24	[91]
				*n*-Hexane	L3	165.10	297.70	24	[91]
				Ethyl acetate	L3	154.88	288.61	24	[91]
				Methanol	L3	127.64	245.73	24	[91]
				Chloroform	L3	146.28	274.42	24	[91]
*Ervatamia coronaria* (Jacq.) Stapf.	Apocynaceae	India	Leaves	Ethyl acetate	L3	97.53	179.37	24	[85]
				Benzene	L3	89.59	166.04	24	[85]
*Eupatorium odoratum* L.	Asteraceae	India	Leaves	Petroleum ether	L4	155.0	290.0	ND	[83]
*Euphorbia hirta* L.	Euphorbiaceae	India	Leaves	Petroleum ether	L4	272.36	703.76	24	[92]
*Euphorbia tirucalli* L.	Euphorbiaceae	India	Stem bark	Petroleum ether	L4	4.25	13.14	24	[92]
*Ficus benghalensis* L.	Moraceae	India	Leaves	Methanol	L2	56.54	109.29	24	[93]
				Methanol	L3	70.29	137.23	24	[93]
				Methanol	L4	80.85	169.58	24	[93]
				Benzene	L2	108.95	227.13	24	[93]
				Benzene	L3	116.09	235.54	24	[93]
				Benzene	L4	159.15	430.91	24	[93]
				Acetone	L2	189.11	444.42	24	[93]
				Acetone	L3	244.41	573.52	24	[93]
				Acetone	L4	288.10	668.71	24	[93]
*Gardenia ternifolia* Schumach. & Thonn.	Rubiaceae	Kenya	Leaves	Acetone	L2	83.31	ND	24	[94]
				Methanol	L2	32.01	ND	24	[94]
*Helicteres velutina* K. Schum.	Malvaceae	Brazil	Root	Ethanol 90%	L4	171.68	403.61	48	[95]
			Stem	Ethanol 90%	L4	138.90	319.37	48	[95]
*Heracleum rigens* Wall.	Apiaceae	India	Seed	Petroleum ether	L2	40.64	65.49	24	[96]
				Chloroform	L2	69.22	132.95	24	[96]
				Ethyl acetate	L2	70.65	93.11	24	[96]
				Methanol	L2	74.70	135.07	24	[96]
				Acetone	L2	97.07	198.99	24	[96]
				Petroleum ether	L3	91.55	162.09	24	[96]
				Chloroform	L3	114.25	179.99	24	[96]
				Ethyl acetate	L3	143.48	207.45	24	[96]
				Methanol	L3	195.57	348.56	24	[96]
				Acetone	L3	234.77	ND	24	[96]
				Petroleum ether	L4	113.69	171.12	24	[96]
				Chloroform	L4	144.64	209.16	24	[96]
				Ethyl acetate	L4	165.43	230.21	24	[96]
				Methanol	L4	231.26	361.68	24	[96]
				Acetone	L4	308.65	577.14	24	[96]
*Hypericum japonicum* Thunb.	Hypericaceae	India	Whole plant	Acetone	L4	13.15	18.54	24	[97]
				n-Hexane	L4	9.63	19.53	24	[97]
				Petroleum ether	L4	8.27	15.2	24	[97]
				Methanol	L4	7.37	11.59	24	[97]
*Jatropha curcas* L.	Euphorbiaceae	India	Leaves	Petroleum ether	L4	8.79	35.39	24	[92]
		Indonesia	Root	Ethanol	ND	44.75	ND	24	[98]
*Limonia acidíssima* L.	Rutaceae	India	Leaves	*n*-Hexane—Purified fractions	L3	4.11 to 23.53	ND	24	[99]
*Lonchocarpus urucu* Killip & A.C. Sm.	Fabaceae	Brazil	Medulla Root	Methanol	L4	33.32	83.69	24	[100]
			Bark Root	Methanol	L4	17.60	55.40	24	[100]
*Maytenus oblongata* Reissek	Celasteraceae	French Guiana	Bark	Ethyl acetate	L3/L4	74.40	ND	24	[101]
*Millettia pachycarpa* Benth.	Fabaceae	India	Root	Ethanol	L3	98.47	ND	24	[102]
*Mirabilis jalapa* L.	Nyctaginaceae	India	Leaves	Benzene	L3	97.03	172.15	24	[103]
				Chloroform	L3	88.20	162.16	24	[103]
				Ethyl acetate	L3	72.77	127.91	24	[103]
				Methanol	L3	64.58	120.28	24	[103]
*Momordica charantia* L.	Cucurbitaceae	India	Leaves	Methanol	L4	199.14	780.10	24	[88]
*Myristica fragans* Houtt.	Myristicaceae	India	Leaves	Methanol	L1	162.03	502.04	24	[53]
				Methanol	L2	194.11	542.56	24	[53]
				Methanol	L3	240.10	604.78	24	[53]
				Methanol	L4	273.90	660.96	24	[53]
				Methanol + ZnO NP	L1	3.44	18.35	24	[53]
				Methanol + ZnO NP	L2	5.25	30.37	24	[53]
				Methanol + ZnO NP	L3	8.02	39.14	24	[53]
				Methanol + ZnO NP	L4	10.28	44.07	24	[53]
*Myristica fragrans* Houtt.	Myristicaceae	Thailand	Flowers	Ethanol	L4 (p-s)	75.45	123.60	24	[60]
*Nerine sarniensis* (L.) Herb.	Amaryllidaceae	South Africa	Bulb	Ethyl acetate	L1	8.0	ND	24	[104]
*Nyctanthes arbor-tristis* L.	Oleaceae	India	Leaves	Petroleum ether	L4	180.0	340.0		[83]
*Ocimum sanctum* L.	Labiatae	India	Leaves and Flowers	Acetone	L4	425.94	ND	24	[105]
				Chloroform	L4	150.40	ND	24	[105]
				Ethyl acetate	L4	350.78	ND	24	[105]
				*n*-Hexane	L4	575.26	ND	24	[105]
				Methanol	L4	175.67	ND	24	[105]
*Ormosia arborea* Vell	Fabaceae	Brazil	Leaves	Ethanol	L3	238.0	347.0	24	[106]
			Seeds	Ethanol	L3	111.0	194.0	24	[106]
*Orthosiphon thymiflorus* (Roth) Sleesen	Labiatae	India	Leaves	n-Hexane	L3	228.13	526.12	24	[107]
				Chloroform	L3	209.72	502.84	24	[107]
				Ethyl acetate	L3	183.35	463.35	24	[107]
				Acetone	L3	163.55	442.32	24	[107]
				Methanol	L3	149.96	426.16	24	[107]
*Pedilanthus tithymaloides* (L.) Poit.	Euphorbiaceae	India	Leaves	Petroleum ether	L4	55.26	256.77	24	[92]
*Pemphis acidula* J.R. Forst. & G. Forst.	Lythraceae	India	Leaves	Methanol	L3	22.10	43.71	24	[108]
				Benzene	L3	43.99	84.87	24	[108]
				Acetone	L3	57.66	106.51	24	[108]
*Phyllanthus amarus* Schumach. & Thonn.	Euphorbiaceae	India	Leaves	Petroleum ether	L4	90.92	384.19	24	[92]
*Piper aduncum* L.	Piperaceae	Brazil	Leaves	*n*-Hexane	L3	342.0	473.0	24	[106]
				Chloroform	L3	192.0	346.0	24	[106]
*Piper hispidum* Sw.	Piperaceae	Brazil	Leaves	Ethanol	L3	169.0	474.0	24	[106]
				Chloroform	L3	567.0	1003.0	24	[106]
*Piper longum* L.	Piperaceae	Thailand	Fruits	Ethanol	L4	2.23	ND	24	[109]
*Piper nigrum* L.	Piperaceae	Philippines	Peppercorns	Ethanol	L3/L4	71.25	9.37	24	[110]
				Ethanol—Fraction 1A	L3/L4	17.10	3.84	24	[110]
				Ethanol—Fraction 1B	L3/L4	18.10	3.84	24	[110]
*Piper ribesoides* Wall.	Piperaceae	Thailand	Wood	Ethanol	L4	8.13	ND	24	[109]
*Piper sarmentosum* Roxb. ex Hunt.	Piperaceae	Thailand	Whole plant	Ethanol	L4	4.06	ND	24	[109]
*Pithecellobium Dulce* (Roxb.) Benth.	Fabaceae	India	Leaves	Methanol	L3	155.78	279.73	24	[111]
				Ethyl acetate	L3	162.36	283.43	24	[111]
				Chloroform	L3	169.08	293.17	24	[111]
				Benzene	L3	176.02	308.88	24	[111]
				n-Hexane	L3	185.14	316.46	24	[111]
			Seeds	Methanol	L3	193.66	377.39	24	[111]
				Ethyl acetate	L3	215.63	416.51	24	[111]
				Chloroform	L3	240.39	461.28	24	[111]
				Benzene	L3	259.42	489.41	24	[111]
				n-Hexane	L3	281.18	516.33	24	[111]
*Scoparia dulcis* L.	Plantaginaceae	Brazil	Leaves	Ethanol 90%	L4	83.43	158.83	48	[112]
*Solanum nigrum* L.	Solanaceae	India	Fruit	Water	L3/L4	359.0	931.0	24	[113]
				n-Hexane	L3/L4	17.63	65.22	24	[113]
*Solanum variabile* Mart.	Solanaceae	Brazil	Leaves	Ethanol	L3	188.0	284.0	24	[106]
*Solanum xanthocarpum* Schrad. & J.C. Wendl.	Solanaceae	India	Fruit	Methanol	L1	170.91	320.62	24	[114]
				Methanol	L2	195.07	366.48	24	[114]
				Methanol	L3	221.45	410.20	24	[114]
				Methanol	L4	253.18	435.16	24	[114]
*Spermacoce latifólia* Aubl.	Rubiaceae	Brazil	Leaves	n-Hexane	L3	415.0	901.0	24	[106]
				Methanol	L3	625.0	1122.0	24	[106]
*Tagetes patula* L.	Asteraceae	Brazil	Seeds	Acetone	L4	15.74	ND	48	[115]
				Ethanol 50%	L4	25.46	ND	48	[115]
*Turnera ulmifolia* L.	Turneracea	Brazil	Leaves	Ethanol	L3	242.0	899.0	24	[106]
*Valeriana hardwickii* Wall.	Valerianaceae	India	Leaves	Petroleum ether	L4	235.0	415.0	ND	[83]
*Ventilago madraspatana* Gaertn.	Rhammnaceae	India	Leaves	Water + AgNP	L3	26.92	ND	24	[112]
				Water	L3	267.27	ND	24	[112]
*Zeuxine gracilis* (Berda) Bl.	Orchidaceae	India	Leaves	Water + AgNP	L3	10.39	23.58	24	[116]

LC_50_ lethal concentration required to kill 50% of the larval population, LC_90_ lethal concentration required to kill 90% of the larval population, ND not described, p-s pyrethroid-susceptible, AgNP silver nanoparticle, ZnONP zinc oxide nanoparticle.

**Table 4 molecules-25-03484-t004:** Adulticidal, pupicidal, ovicidal, repellent and oviposition activities of organic/aqueous extracts against the *Ae. aegypti* mosquito.

Plant Species	Family	Country	Part Used	Extraction Solvent	Activity	Results	Time (h)	Reference
*Alpinia purpurata* (Viell.) K. Schum.	Zingiberaceae	Brazil	Red Flowers	Water	Oviposition	Oviposition disruptive effect	24	[67]
			Pink Flowers	Water	Oviposition	Oviposition disruptive effect	24	[67]
*Aristolochia bracteata* Retz.	Aristolochiaceae	India	Leaves	Methanol	Ovicide	Zero hatchability at 240 ppm	48	[81]
				Methanol	Repellent	100% of repellency at 6 mg/cm^2^	3	[81]
*Artemisia herba-alba* Asso	Asteraceae	Saudi Arabia	Leaves	Water	Adulticide—strain from India	LC_50_ 327.15 µg/mL LC_90_ = 779.98 µg/mL	24	[82]
				Water	Adulticide—strain from Saudi Arabia	LC_50_ 450.21 µg/mL LC_90_ 1153.18 µg/mL	24	[82]
				Water + AgNP	Adulticide—strain from India	LC_50_ 8.71 µg/mL LC_90_ 39.88 µg/mL	24	[82]
				Water + AgNP	Adulticide—strain from Saudi Arabia	LC_50_ 25.62 µg/mL LC_90_ 48.88 µg/mL	24	[82]
*Buddleja polystachya* Fresen.	Buddlejaceae	Saudi Arabia	Flowers	n-Hexane	Adulticide	96.7% mortality at 5 µg/mg female	ND	[84]
				Ethanol	Adulticide	83.3% mortality at 5 µg/mg female	ND	[84]
			Aerial parts	*n*-Hexane	Adulticide	100% mortality at 5 µg/mg female	ND	[84]
				Ethanol	Adulticide	90% mortality at 5 µg/mg female	ND	[84]
*Caesalpinia pulcherrima* (L.) Sw.	Fabaceae	India	Leaves	Methanol	Ovicide	Zero hatchability at 300 ppm	48	[85,117]
				Methanol	Repellent	100% of repellency at 5 mg/cm^2^	3	[85,114]
				Ethyl acetate	Ovicide	Zero hatchability at 450 ppm	48	[85,117]
				Ethyl acetate	Repellent	100% of repellency at 5 mg/cm^2^	1.5	[85,117]
				Benzene	Ovicide	Zero hatchability at 375 ppm	48	[85,117]
				Benzene	Repellent	100% of repellency at 5 mg/cm^2^	2	[85,117]
*Cardiospermum halicacabum L.*	Sapindaceae	India	Leaves	Methanol	Repellent	100% of repellency at 5 mg/cm^2^	3	[118]
				*n*-Hexane	Repellent	100% of repellency at 5 mg/cm^2^	3	[118]
				Ethyl acetate	Repellent	100% of repellency at 5 mg/cm^2^	3	[118]
				Chloroform	Repellent	100% of repellency at 5 mg/cm^2^	3	[118]
				Benzene	Repellent	100% of repellency at 5 mg/cm^2^	3	[118]
*Cassia fistula* L.	Fabaceae	India	Leaves	Methanol	Ovicide	Zero hatchability at 120 ppm	48	[86]
				Methanol	Repellent	100% of repellency at 5 mg/cm^2^	6	[86]
				Benzene	Ovicide	Zero hatchability at 140 ppm	48	[86]
				Benzene	Repellent	100% of repellency at 5 mg/cm^2^	5	[86]
				Acetone	Ovicide	Zero hatchability at 160 ppm	48	[86]
				Acetone	Repellent	100% of repellency at 5 mg/cm^2^	4.3	[86]
*Cinnamosma fragrans* Baill.	Canellaceae	Madagascar	Root barks	Methanol	Adulticide	LC_50_ 0.17 µg/mg female	24	[87]
				Methanol	Repellent	80% of repellency at 20.8 µg/cm^2^	3	[87]
*Coccinia indica* Wight & Arn.	Cucurbitaceae	India	Leaves	Benzene	Ovicide	Zero hatchability at 250 ppm	48	[119]
				Benzene	Repellent	100% of repellency at 5 mg/cm^2^	3	[119]
				*n*-Hexane	Ovicide	Zero hatchability at 300 ppm	48	[119]
				*n*-Hexane	Repellent	100% of repellency at 1 mg/cm^2^	2.5	[119]
				Ethyl acetate	Ovicide	Zero hatchability at 250 ppm	48	[119]
				Ethyl acetate	Repellent	100% of repellency at 2.5 mg/cm^2^	2.5	[119]
				Methanol	Ovicide	Zero hatchability at 200 ppm	48	[119]
				Methanol	Repellent	100% of repellency at 5 mg/cm^2^	3.5	[119]
				Chloroform	Ovicide	Zero hatchability at 250 ppm	48	[119]
				Chloroform	Repellent	100% of repellency at 2.5 mg/cm^2^	2.5	[119]
*Eclipta alba* (L.) Hassk	Asteraceae	India	Leaves	Benzene	Ovicide	Zero hatchability at 350 ppm	48	[119]
				*n*-Hexane	Ovicide	21% hatchability at 350 ppm	48	[119]
				Ethyl acetate	Ovicide	Zero hatchability at 350 ppm	48	[119]
				Methanol	Ovicide	Zero hatchability at 300 ppm	48	[91]
				Chloroform	Ovicide	Zero hatchability at 350 ppm	48	[119]
*Ervatamia coronaria* (Jacq.) Stapf.	Apocynaceae	India	Leaves	Methanol	Ovicide	Zero hatchability at 200 ppm	48	[117]
				Methanol	Repellent	100% of repellency at 5 mg/cm^2^	3	[117]
				Ethyl acetate	Ovicide	Zero hatchability at 300 ppm	48	[117]
				Ethyl acetate	Repellent	100% of repellency at 5 mg/cm^2^	2	[117]
				Benzene	Ovicide	Zero hatchability at 250 ppm	48	[117]
				Benzene	Repellent	100% of repellency at 5 mg/cm^2^	2.5	[117]
*Limonia acidíssima* L.	Rutaceae	India	Leaves	*n*-Hexane—Purified fractions	Ovicide	78.4 hatchability at 10 ppm	120	[99]
				n-Hexane—Purified fractions	Pupicide	LC_50_ 4.19—39.48 µg/mL	24	[99]
*Mentha piperita* L.	Lamiaceae	India	Whole plant	Methanol	Repellent	Repellency	ND	[120]
*Millettia pachycarpa* Benth.	Fabaceae	India	Root	Ethanol	Ovicide	Zero hatchability at 200 ppm	24	[102]
*Myristica fragans* Houtt.	Myristicaceae	India	Leaves	Methanol	Pupicide	LC_50_ 359.08 µg/mL LC_90_ 803.52 µg/mL	24	[53]
				Methanol + ZnONP	Pupicide	LC_50_ 14.63 µg/mL LC_90_ 51.22 µg/mL	24	[53]
				Methanol	Adulticide	LC_50_ 180.26 µg/mL LC_90_ 368.93 µg/mL	24	[53]
				Methanol + ZnONP	Adulticide	LC_50_ 15.0 µg/mL LC_90_ 34.2 µg/mL	24	[53]
*Nerine sarniensis* (L.) Herb.	Amaryllidaceae	South Africa	Bulbs	Ehtyl acetate	Adulticide	LC_50_ 4.6 µg/mg female	24	[104]
*Parthenium hysterophorus*	Asteraceae	India	Leaves	Ether	Repellent	99.6% of repellency at 1000 ppm	48	[121]
				Ether	Ovicide	Zero hatchability at 1000 ppm	48	[121]
				Benzene	Repellent	93.8% of repellency at 1000 ppm	48	[121]
				Benzene	Ovicide	Zero hatchability at 1000 ppm	48	[121]
*Pemphis acidula* J.R. Forst. & G. Forst.	Lythraceae	India	Leaves	Methanol	Ovicide	Zero hatchability at 450 ppm	48	[108]
				Acetone	Ovicide	Zero hatchability at 500 ppm	48	[108]
*Pithecellobium Dulce* (Roxb.) Benth.	Fabaceae	India	Leaves	Methanol	Ovicide	Zero hatchability at 400 ppm	48	[111]
				Ehtyl acetate	Ovicide	Zero hatchability at 500 ppm	48	[111]
				Chloroform	Ovicide	Zero hatchability at 500 ppm	48	[111]
				Benzene	Ovicide	Zero hatchability at 600 ppm	48	[111]
				*n*-Hexane	Ovicide	Zero hatchability at 600 ppm	48	[111]
			Seeds	Methanol	Ovicide	Zero hatchability at 625 ppm	48	[111]
				Ethyl acetate	Ovicide	Zero hatchability at 750 ppm	48	[111]
				Chloroform	Ovicide	Zero hatchability at 750 ppm	48	[111]
				Benzene	Ovicide	Zero hatchability at 750 ppm	48	[111]
*Solanum xanthocarpum* Schrad. & J.C. Wendl.	Solanaceae	India	Fruit	Methanol	Pupicide	LC_50_ 279.52 µg/mL LC_90_ 462.10 µg/mL	24	[114]
*Ventilago madraspatana* Gaertn.	Rhammnaceae	India	Leaves	Water + AgNP	Ovicide	Zero hatchability at 120 ppm	48	[112]
				Water	Ovicide	Zero hatchability at 400 ppm	48	[112]
				Water + AgNP	Adulticide	LC_50_ 44.85 µg/mL	24	[112]
				Water	Adulticide	LC_50_ 334.46 µg/mL	24	[112]
*Zeuxine gracilis* (Berda) Bl.	Orchidaceae	India	Leaves	Water + AgNP	Ovicide	Zero hatchability at 12 ppm	48	[116]
					Adulticide	LC_50_ 27.90 µg/mL LC_90_ = 59.20 µg/mL	24	[116]

LC_50_ lethal concentration required to to kill 50% of the mosquito population, LC_90_ lethal concentration required to kill 90% of the mosquito population, ND not described, AgNP silver nanoparticle, ZnONP zinc oxide nanoparticle.

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
