# Peer review of "Plant Natural Products for the Control of Aedes aegypti: The Main Vector of Important Arboviruses"

_molecules, 2020, doi:10.3390/molecules25153484_

Round 1
Reviewer 1 Report
This is a very ambitious review covering an enormous amount of material. The review is very thorough and provides excellent coverage of a complicated topic. The authors have done a superb job with this review and my comments are only editorial. The biggest problem is that the writing style seems disruptive and choppy because of the many small 1-2 sentence paragraphs. Each of these paragraphs related to a different plant, however with so many paragraphs, it is disruptive. Some of the sentences can easily be incorporated into the preceding or following paragraphs. There are too many to individually correct but this does should be addressed.
Line 27 convention is Aedes aegypti (Diptera: Culicidae). You should not have to explain that.
Line 47-48 need a reference for those transmission routes
Line 52 remove the first ‘the’
Figure 1 has yellow fever indicated but yellow fever is not mentioned in the text. Either add something about yellow fever or remove it from the figure.
Line 84-90 This is your paragraph on toxic synthetic insecticides and I think their use against adult mosquitoes is underrepresented. The use of foggers and aerial applications against adults was and is pervasive and the cause of many of the resistance problems. Addition of a sentence or so would help
Line 92 it is not application of bacteria, it is incorporation of highly specific strains of Wolbachia
Line 94 not any bacteria, it is only the specific strain of Wolbachia
Line96 use of the word ‘forms’ is not conventional, you can use ‘stages’ instead
Line 137 minimal residues. I am not sure what you mean, that they break down, that less material is applied?
Line 143-144 reference?
Line 150-158 references?
Line 166 Some essential oils are considered moderately toxic to mammals and some such as methyl eugenol have human risk. These risks are provided by material safety data sheets accompanying neat compounds when purchased. Purity and concentration are critical aspects so a global statement saying that they are safe is not quite accurate. Perhaps adjust your statement a bit
Line 166-167 This seems like an oversimplification of some data. Depends on the compound and purity.
Line181-183 what are you saying? Reference? Evidence?
Line 184 were used for what, in what study, no reference.
Line 188 plant species? Reword to “highlights 14 plant species with very active essential oils” the oil are active not the plants. Do the same for the rest of the sentence.
Line 192 – 282 this is a good section but I think that it needs a bit of organization. I would list the tables and figures first and that in itself will help a lot with the flow as the reader can follow through the tables. I think that you need to start a paragraph with an overview statement then include your sentences which have the information. Right now it is a lot of text that seems disconnected and an overview would help it pull together. You state that there are 14 very active essential oils, are these the ones that you are mentioning in the text? After them, then give an overview of the next group. By grouping these sentences together it will really help the reader see what you are saying. This type of organization is also needed for the other sections where the reader can get lost with the many separate paragraphs because they are not seeing the larger picture. Grouping the sentences together with an initial sentence that gives an overview of what you are talking about will help at lot. This is information dense material so the overviews or conclusion sentences will really help. It may be difficult to group some sentences as the topic was differ a lot, however even if some of the sentences are grouped, I think that it will enhance the readability of the manuscript.
Line 215 what references?
Line 460-463 I would move this to the top of the section so that the reader will look at the figures while reading through the text.
Line 781 articles or studies?
Line 826 remove ‘s’ from extracts
Line 828 comprise or provide?
Author Response
Dear Editor.
Please find below the answers to the comments raised by the Reviewer 1. All the corrections are highlighted in the text of the manuscript
Line 27 convention is Aedes aegypti (Diptera: Culicidae). You should not have to explain that.
- Suggestion accepted. Phrase rewritten.
Line 47-48 need a reference for those transmission routes.
- Reference inserted.
Line 52 remove the first ‘the’.
- Done.
Figure 1 has yellow fever indicated but yellow fever is not mentioned in the text. Either add something about yellow fever or remove it from the figure.
- We believe we can keep yellow fever once the vector is the same for the other diseases what is also mentioned in the text line 39.
Line 84-90 This is your paragraph on toxic synthetic insecticides and I think their use against adult mosquitoes is underrepresented. The use of foggers and aerial applications
against adults was and is pervasive and the cause of many of the resistance problems. Addition of a sentence or so would help.
- Accepted. A phrase was added and a more detailed discussion is shown below in the text.
Line 92 it is not application of bacteria, it is incorporation of highly specific strains of Wolbachia.
- Accepted according suggestion.
Line 94 not any bacteria, it is only the specific strain of Wolbachia. .
- Accepted according suggestion
Line96 use of the word ‘forms’ is not conventional, you can use ‘stages’ instead.
- Done as suggested
Line 137 minimal residues. I am not sure what you mean, that they break down, that less material is applied?
- This refers to minimal residues of degradation that occur by natural mechanisms, causing lower impact to the environment.
Line 143-144 reference?
- Inserted.
Line 150-158 references?
- In this paragraph we just discuss what the review is about therefore we do not believe there is a need for references. They will show up in right place throughout the text.
Line 166 Some essential oils are considered moderately toxicto mammals and some such as methyl eugenol have human risk. These risks are provided by material safety data sheets accompanying neat compounds when purchased. Purity and concentration are critical aspects so a global statement saying that they are safe is not quite accurate. Perhaps adjust your statement a bit.
- Paragraph re-written as suggested
Line 166-167 This seems like an oversimplification of some data. Depends on the compound and purity.
- Phrase was re-written and the values of toxicity was kept once this is the way they usually appear in the papers dealing with insecticidal activities.
Line181-183 what are you saying? Reference? Evidence?
- Paragraph removed.
Line 184 were used for what, in what study, no reference.
- Phrase re-written.
Line 188 plant species? Reword to “highlights 14 plant species with very active essential oils” the oil are active not the plants. Do the same for the rest of the sentence.
- Done.
Line 192 – 282 this is a good section but I think that it needs a bit of organization. I would list the tables and figures first and that in itself will help a lot with the flow as the reader can follow through the tables. I think that you need to start a paragraph with an overview statement then include your sentences which have the information. Right now it is a lot of text that seems disconnected and an overview would help it pull together. You state that there are 14 very active essential oils, are these the ones that you are mentioning in the text?
After them, then give an overview of the next group. By grouping these sentences together it will really help the reader see what you are saying. This type of organization is also needed for the other sections where the reader can get lost with the many separate paragraphs because they are not seeing the larger picture. Grouping the sentences together with an initial sentence that gives an overview of what you are talking about will help at lot. This is information dense material so the overviews or conclusion sentences will really help. It may be difficult to group some sentences as the topic was differ a lot, however even if some of the sentences are grouped, I think that it will enhance the readability of the manuscript.
- We have followed the suggestion of condensing the paragraphs, and are now organized according to the larvicidal activity. We repositioned tables and figures as suggested.
Line 215 what references?
- Inserted.
Line 460-463 I would move this to the top of the section so that the reader will look at the figures while reading through the text.
- Done.
Line 781 articles or studies?
- Used Studies.
Line 826 remove ‘s’ from extracts.
- Done.
Line 828 comprise or provide?
- Used Provide.
Reviewer 2 Report
This article present study to review details plant species and their secondary metabolites that have demonstrated insecticidal properties (ovicidal, larvicidal, pupicidal, adulticidal, repellent and ovipositional effects) against the mosquito, together with their mechanisms of action. This review is interesting. I think this is a highly value of paper for reader's reference. I recommend Molecules accept this paper to publish after minor revision. However, there are several points which have to be carefully considered by the authors for its possible publication as follows:
1. P37, Line 597, ” n-isobutylamide” should be changed as “N-isobutylamide”.
Author Response
Dear Editor.
Please find below the answers to the comments raised by the Reviewer 2. All the corrections are highlighted in the text of the manuscript
Everywhere in the text:
- the names of the plants should be written correctly: the names of the authors are missing.
- Done.
- n-hexane should be written instead of hexane.
- Done.
In: Introduction
- line 151: “…by steam distillation or…” should be written instead of “… by steam extraction or…”.
- Done.
- line 193: hexadecane (1) and heptacosane (2) are not fatty acid derivatives. This compounds are saturated aliphatic hydrocarbons.
- Changed according suggestion
- line 229: Lamiaceae should be written instead of Lamiceae.
- Done.
- line 236: It should be described what the symbol “~“ stands for.
- Changed for ca.
- line 282: The sentence “The chemical structures of …” should be written before the line 192.
- Done.
- in Figure 3: p-cymene is not terpene!! p-Cymene is hydrocarbon phenylpropanoids.
- We are afraid to say that we should keep p-cymene as terpene and not phenylpropanoid a suggested by the reviewer
- line 463: The sentence “Figure 4 details…” should be written before the line 309.
- Done.
- line 374: … silver nano particle… should be written instead of …AgNP…, because the list of the abbreviations are missing.
- Done.
- line 378: …zinc oxide nano particle… should be written instead of …ZnO NP…, because the list of the abbreviations are missing.
- Done.
- in Figure 4: Phenylpropanoids should be written instead of “others”.
- We have not changed even though we agree that dihydrocinnamic acid is a real phenylpropanoid however 2,4,-diterbutyltphenol is not.
- line 547: p-cymene(17) is not terpene!!
- The same as above
- line 582: The sentence “The chemical structure…” should be written before the line 569.
- Done.
- line 608: The sentence “The chemical structure…” should be written before the line 588.
- Done.
- line 537: The sentence “Thiophene and fatty acid…” should be written before the line 612.
- Done.
- line 634: The sentence “The corresponding chemical…” should be written before the line 631.
- Done.
- line 642: The sentence “Figure 12…” should be written before the line 638.
- Done.
- line 658: The sentence “The chemical structures…” should be written before the line 648.
- Done.
- line 673: The sentence “The chemical structures…” should be written before the line 662.
- Done.
- In my opinion the text should be reduced and all Tables should be excluded, because the respective data are described in the text.
- We have revised the text and we believe the tables should be kept once they contain useful information that are not in the text.
In: Conclusions
- some corrections of the text are necessary, because it is not clear enough now.
- We have changed some points and believe that is better now
In: References
- the list of references should be written according to the guideline for authors, for example № 3.
- Done. All references were rechecked and corrected accordingly.
- the references should be correctly written, for example № 112, 138.
- Done.
Reviewer 3 Report
Everywhere in the text:
- the names of the plants should be written correctly: the names of the authors are missing.
- n-hexane should be written instead of hexane.
In: Introduction
- line 151: “…by steam distillation or…” should be written instead of “… by steam extraction or…”.
- line 193: hexadecane (1) and heptacosane (2) are not fatty acid derivatives. This compounds are saturated aliphatic hydrocarbons.
- line 229: Lamiaceae should be written instead of Lamiceae.
- line 236: It should be described what the symbol “~“ stands for.
- line 282: The sentence “The chemical structures of …” should be written before the line 192.
- in Figure 3: p-cymene is not terpene!! p-Cymene is hydrocarbon phenylpropanoids.
- line 463: The sentence “Figure 4 details…” should be written before the line 309.
- line 374: … silver nano particle… should be written instead of …AgNP…, because the list of the abbreviations are missing.
- line 378: …zinc oxide nano particle… should be written instead of …ZnO NP…, because the list of the abbreviations are missing.
- in Figure 4: Phenylpropanoids should be written instead of “others”.
- line 547: p-cymene(17) is not terpene!!
- line 582: The sentence “The chemical structure…” should be written before the line 569.
- line 608: The sentence “The chemical structure…” should be written before the line 588.
- line 537: The sentence “Thiophene and fatty acid…” should be written before the line 612.
- line 634: The sentence “The corresponding chemical…” should be written before the line 631.
- line 642: The sentence “Figure 12…” should be written before the line 638.
- line 658: The sentence “The chemical structures…” should be written before the line 648.
- line 673: The sentence “The chemical structures…” should be written before the line 662.
- In my opinion the text should be reduced and all Tables should be excluded, because the respective data are described in the text.
In: Conclusions
- some corrections of the text are necessary, because it is not clear enough now.
In: References
- the list of references should be written according to the guideline for authors, for example № 3.
- the references should be correctly written, for example № 112, 138.
Author Response
Dear Editor.
Please find below the answers to the comments raised by the Reviewer 2. All the corrections are highlighted in the text of the manuscript
1. P37, Line 597, ” n-isobutylamide” should be changed as “N-isobutylamide”.
R: Done.
Round 2
Reviewer 3 Report
c
In: Introduction
- in Figure 3: p-cymene (17) is not terpene!! p-Cymene is hydrocarbon phenylpropanoids.
In: Conclusions
- some corrections of the text are necessary, because it is not clear enough now.
Author Response
In: Introduction
in Figure 3: p-cymene (17) is not terpene!! p-Cymene is hydrocarbon phenylpropanoids.
R: Regarding the consideration of p-cymene to be or not a terpene we believe we should keep it as terpene even though the reviewer does not agree with us therefore we have not taken into account his/her suggestion. It does not seem that p-cymene should be considered a hydrocarbon phenylpropanoid.
Our points are: p-cymene is a terpene according to Poulose and Croteau, 1978 (Biosynthesis of aromatic monoterpenes conversion of y-terpinene to ρ-cymene and thymol in Thymus vulgaris L. Arch Biochem Biophys. 1978, 187, 307-314.). Also in the book Medicinal Natural Products, by P.M. Dewick, (John Wiley & Sons Ltd, 2002; pp. 178, 2a. ed), p-cymene is considered a terpene. Dewick states “p-cymene, and the phenol derivatives thymol and carvacrol found in thyme (Thymus vulgaris; Labiatae/Lamiaceae), are representatives of a small group of aromatic compounds that are produced in nature from isoprene units, rather than by the much more common routes to aromatics involving acetate or shikimate”. Additionally, Howyzeh et al., 2020 (Comparative transcriptome analysis to identify putative genes involved in thymol biosynthesis pathway in medicinal plant Trachyspermum ammi L. Sci Rep. 2020, 8, 13405) studied the genes involved in biosynthesis of thymol, a phenol derivative of p-cymene. In this study was identified the genes encoding enzymes for the intermediate stages of terpenoid biosynthesis pathways in medicinal plant Trachyspermum ammi L.
Finally, it seems to us contradictory to have in the same journal different denominations for the same compound. Having a quick look in the papers published by the journal Molecules, we could easily found that p-cymene is referred as terpene. We certainly have not gone through all of them, so the four following are a sample of this observation.
- Molecules2020, 25(1), 148; https://doi.org/10.3390/molecules25010148;
- Molecules2020, 25(4), 827; https://doi.org/10.3390/molecules25040827;
- Molecules2020, 25(4), 804; https://doi.org/10.3390/molecules25040804;
- Molecules2020, 25(5), 1100; https://doi.org/10.3390/molecules25051100
We hope that now is clear why we are keeping p-cymene terpene. We also believe that this is the more appropriate denomination for it.
In: Conclusions
some corrections of the text are necessary, because it is not clear enough now.
R: Section re-written.